# MASKED DUAL-TEMPORAL AUTOENCODERS FOR SEMI-SUPERVISED TIME-SERIES CLASSIFICATION

## ABSTRACT

In this study, we propose a novel framework for semi-supervised time-series classification based on masked time-series modeling, a recent advance in self-supervised learning effective for capturing intricate temporal structures within time series. The proposed method effectively extracts intrinsic semantic information from unlabeled instances by reflecting diverse temporal resolutions and considering various masking ratios during model training. Then, we incorporate the semantic information extracted from unlabeled time series with supervisory features, including hard-to-learn class information, learned from labeled ones to improve classification performance. Through extensive experiments on semi-supervised time-series classification, we demonstrate the superiority of our approach by achieving state-of-the-art performance.

## 1  INTRODUCTION

Recent advances in deep learning have shown promising performance in time-series classification, a fundamental task driven by the growing accessibility of vast time-series data (Bagnall et al., 2017; Lee et al., 2023). These remarkable achievements require numerous labeled training instances, but in practice, they are frequently lacking, whereas unlabeled ones abound. Annotating all unlabeled time series within a reasonable time and cost is often infeasible; thereby, semi-supervised learning, which leverages both labeled and unlabeled instances to mitigate label sparsity, has attracted considerable attention in time-series classification (de Carvalho Pagliosa & de Mello, 2018).

The semi-supervised learning aims to enhance the generalization capability of models by incorporating a large set of unlabeled instances with a few labeled ones during model training, leading to boosting model performance. Recent studies for semi-supervised learning have actively exploited self-supervised learning, e.g., contrastive learning, to learn implicit structures within unlabeled time series under the supervision of self-generated labels (Liu et al., 2022; 2023). However, they have two limitations. First, most of them capture coarse-grained context information focused on instance-level, insufficiently recognizing temporal patterns of time series (Wang & Isola, 2020). Second, the model performance highly depends on techniques to construct self-generated labels, such as data augmentations and time sampling functions (You et al., 2021). Especially for time series, it is also challenging to adopt proper perturbations that do not corrupt the time-series nature (Yue et al., 2022).

The concept of masked modeling has emerged in natural language processing and computer vision to deal with high sensitivity to constructing self-generated labels and to capture fine-grained context information (Devlin et al., 2018; Xie et al., 2022). The masked modeling aims to learn useful representations reflecting semantic information from data by enabling a model to reconstruct the masked content based on the unmasked part. However, unlike texts and images, which possess rich semantic information within words or patches, the semantic information of time series is generally contained in temporal variations, such as trend and periodicity (Dong et al., 2023). Thus, to sufficiently consider time-series characteristics, some studies extended this concept to time series, which is called *masked time-series modeling (MTM)* (Nie et al., 2022; Dong et al., 2023). Despite its empirical success, MTM has not yet been introduced to semi-supervised time-series classification.

Moreover, directly applying the existing MTMs to semi-supervised time-series classification still has two potential drawbacks. First, it is challenging to reflect diverse temporal resolutions. Considering them can enrich the semantic information of time series because its temporal dependencies span different time intervals (Zerveas et al., 2021). However, the transformer architecture adopted as

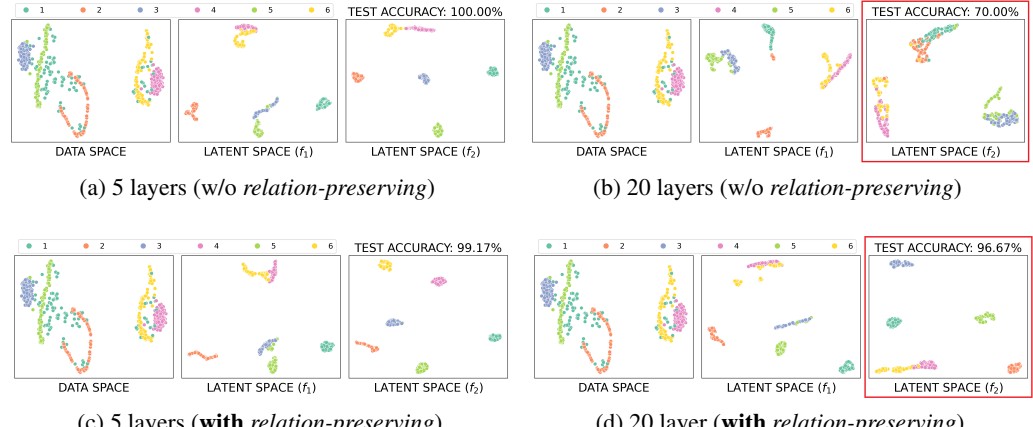

Figure 1: Visualization of data space and latent spaces produced by the sequential sub-encoders, $f_1$ and $f_2$, for *SyntheticControl* dataset. In (a) and (b), we present each space when $f_2$ has 5 and 20 layers, respectively, without *relation-preserving*. In contrast, (c) and (d) show each space when $f_2$ has 5 and 20 layers, respectively, with *relation-preserving*. In (b), representations of each class obtained from $f_2$ exhibit diminished distinctiveness compared to those generated by $f_1$. However, in (d) with *relation-preserving*, representations of each class remain distinguishable.

an encoder in MTMs insufficiently captures temporal patterns at different time scales (Yang & Lu, 2023). Second, they are sensitive to masking ratios; it is often impractical to individually explore the optimal one for each dataset originating from various sources within reasonable time and cost.

To address these challenges, we propose the first MTM-based framework for semi-supervised time-series classification, *masked dual-temporal autoencoders (MDTA)*. MDTA captures relevant semantic information from unlabeled time series and incorporates it with supervisory features obtained from labeled ones to enhance model performance. Specifically, we develop a *dual-temporal encoder* comprising two sequential sub-encoders to learn intrinsic temporal patterns within time series by capturing temporal dependencies at different time scales. However, as shown in Figures 1(a) and (b), our encoder can cause information loss due to its deep architecture, leading to performance degradation. Thus, we introduce a simple yet effective loss function, *relation-preserving*, to ensure a lossless flow of temporal information within the encoder. Moreover, during model training, we use *random masking ratios* to avoid exploring optimal masking ratios while further enhancing the ability to capture temporal relations of time series. Then, by sharing the dual-temporal encoder, MDTA directly classifies labeled instances and follows the masked modeling procedure for unlabeled ones. Through this approach, labeled instances provide useful supervisory features for classification, and unlabeled ones enrich the semantic information of time series, improving classification performance.

The superiority of the proposed method is demonstrated by extensive comparative experiments on semi-supervised time-series classification, where it outperforms state-of-the-art methods (SOTAs) by successfully leveraging semantic information from unlabeled time series.

This study has the following contributions:

- We propose a novel MTM-based framework for semi-supervised time-series classification. To our knowledge, this work is the first exploration of MTM for this purpose.
- To effectively capture intricate temporal patterns within time series across diverse temporal resolutions, we develop a *dual-temporal encoder* comprising two sequential sub-encoders. In addition, we solve the potential information loss problem between the sub-encoders by introducing a *relation-preserving* loss function.
- We use *random masking ratios* at each training epoch to avoid the high-cost tuning process for searching optimal masking ratios along with enhancing classification performance.
- The proposed method captures the inherent temporal information of time series and successfully incorporates them with supervisory features, achieving outstanding performance in semi-supervised time-series classification compared to SOTAs.

## 2 RELATED WORKS

Label sparsity is one of the practical obstacles hindering the use of deep learning in time-series classification. Thus, to alleviate reliance on labeled instances, semi-supervised learning for time-series classification has been studied extensively (de Carvalho Pagliosa & de Mello, 2018).

Some recent studies for semi-supervised time-series classification have exploited self-supervised learning, such as contrastive learning, to extract context information from unlabeled time series. For example, Jawed et al. (2020) proposed a semi-supervised time-series classification method combining self-supervised learning and multi-task learning. Fan et al. (2021) identified temporal relations by using the past-future segments and constructing the positive and negative pairs to extract useful context from unlabeled instances. Extending Fan et al. (2021), Xi et al. (2022) considered temporal patterns between not only the past and future segments but also the present one. In addition, Liu et al. (2022) learned temporal structures of unlabeled instances with self-generated labels obtained by randomly applying time sampling functions to the input time series; Eldele et al. (2023) introduced temporal and contextual contrasting for semi-supervised time-series classification (see Section C). However, these methods have some limitations, such as high sensitivity to constructing self-generated labels and insufficient reflection of context information.

To address these problems, masked modeling has emerged in natural language processing and computer vision (Devlin et al., 2018; Xie et al., 2022), and some studies have recently extended masked modeling to time series. For example, in time-series representation learning, Zerveas et al. (2021) designed a transformer encoder for MTM, while Dong et al. (2023) utilized MTM to capture complementary temporal variations from multiple masked series. In addition, Nie et al. (2022) achieved performance improvement in long-term time-series forecasting using MTM. Despite their empirical success, there have not yet been any attempts to introduce MTM to semi-supervised time-series classification. Moreover, conventional MTMs have two notable drawbacks: the incapability to reflect diverse temporal resolutions and high sensitivity to masking ratios.

In contrast, our method, MDTA, is the first MTM framework for semi-supervised time-series classification. It captures valuable semantic information from unlabeled time series by effectively reflecting diverse temporal resolutions and using random masking ratios during model training. Then, we achieved superior classification performance by incorporating the obtained semantic information with supervisory features learned from labeled instances.

## 3 PROPOSED METHOD

### 3.1 PROBLEM STATEMENT

Let $\mathbb{D} = \{(\boldsymbol{x}_i, y_i)\}_{i=1}^n$ be a set of $n$ samples, where $\boldsymbol{x}_i \in \mathbb{R}^{t \times v}$ is a time-series instance with $t$ lengths and $v$ variables, and $y_i$ denotes the class label of $\boldsymbol{x}_i$. We suppose some of the labels to be missing; thereby, $\mathbb{D}$ is split into two subsets: a labeled set $\mathbb{D}_\ell = \{(\boldsymbol{x}_i, y_i)\}_{i=1}^{n_\ell}$ of size $n_\ell$ and an unlabeled set $\mathbb{D}_u = \{(\boldsymbol{x}_i, \cdot)\}_{i=n_\ell+1}^n$ of size $n_u = n - n_\ell$. We define two sequential sub-encoders $f_1 : \boldsymbol{x} \to \boldsymbol{u}$ and $f_2 : \boldsymbol{u} \to \boldsymbol{z}$, and a decoder $g : \boldsymbol{z} \to \hat{\boldsymbol{u}}$, where $\boldsymbol{z} \in \mathbb{R}^{t \times d_z}$, $\boldsymbol{u} \in \mathbb{R}^{t \times d_u}$, and $\hat{\boldsymbol{u}} \in \mathbb{R}^{t \times d_u}$. Especially for $\mathbb{D}_\ell$, we also define a classification head $h : \boldsymbol{z} \to \hat{y}$. The objective is optimizing $f$, $g$, and $h$ using all accessible instances in $\mathbb{D}$ to improve classification performance.

### 3.2 MASKED DUAL-TEMPORAL AUTOENCODERS

To enhance classification performance by effectively incorporating a large set of unlabeled instances with labeled ones, we propose MDTA, a novel MTM framework for semi-supervised time-series classification. It consists of three components: *dual-temporal encoder*, *simple decoder*, and *classification head*. Figure 2 shows an overview of the proposed method.

#### 3.2.1 DUAL-TEMPORAL ENCODER

To effectively learn inherent temporal structures of time series, we develop a *dual-temporal encoder* $f$ sequentially configured with multi-resolution sub-encoder $f_1$ and transformer-based sub-encoder $f_2$ ($f := f_2 \circ f_1$). Note that the weights of this encoder are shared in MTM and supervised training.

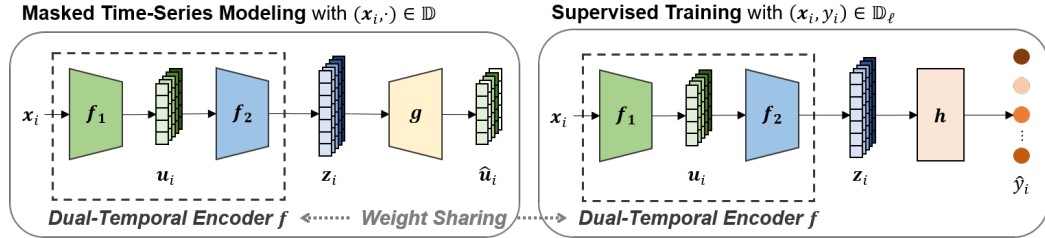

Figure 2: Overview of the proposed method

**Multi-resolution sub-encoder.** Despite the effectiveness and scalability of the transformer used as an encoder in MTMs, it has a limitation in controlling diverse temporal resolutions (Yang & Lu, 2023). Since temporal dependencies span various time intervals, reflecting diverse resolutions can significantly improve model performance for time-series classification (Zerveas et al., 2021). Thus, we construct the multi-resolution sub-encoder $f_1$ before the transformer-based sub-encoder $f_2$ to capture temporal dependencies at different time scales.

Let $\boldsymbol{x}_i \in \mathbb{R}^{t \times v} = [\boldsymbol{x}_{i,1}, \cdots, \boldsymbol{x}_{i,t}]$ be a time series composed of a sequence of $t$ observations. $\boldsymbol{x}_i$ are mapped onto a $d_u$-dimensional latent space along the temporal dimension using $f_1$ as follows:

$$\boldsymbol{u}_i = f_1(\boldsymbol{x}_i), \tag{1}$$

where $\boldsymbol{u}_i \in \mathbb{R}^{t \times d_u} = [\boldsymbol{u}_{i,1}, \cdots, \boldsymbol{u}_{i,t}]$ is the high-level temporal features used as the input for the subsequent sub-encoder $f_2$.

Specifically, the sub-encoder $f_1$ is designed by one-dimensional convolutional layers with causal padding and dilated filters (*DilatedConv*). This architecture ensures that the model does not perturb the temporal order of the input time series and considers diverse temporal resolutions by gradually increasing dilation rates $\rho$. In particular, the causal padding prevents the convolution filter from observing future inputs beyond the current time step by zero-padding the left side of $\boldsymbol{x}_i$. In addition, the dilated filters, convolution filters with strides controlled by $\rho$, allow the model to recognize various temporal patterns at different time scales. For a single time step $\tau$[1], the output of *DilatedConv*, $\boldsymbol{x}'_{i,\tau}$, is calculated by $\boldsymbol{x}'_{i,\tau} = \sum_{\kappa=0}^{k-1} \boldsymbol{x}_{i,(\tau-\kappa\rho-(k-1)\rho)} \times c_\kappa$, where $c_\kappa$ is the weight (or kernel coefficient) at time step $\kappa$ in the convolution filter, and $k$ is the filter size.

By passing $\boldsymbol{x}_i$ through several temporal blocks comprising *DilatedConv*s and GeLU activation functions (see Figure 3(a)), we obtain temporal features $\boldsymbol{u}_i$. By the sub-encoder $f_1$, we learn high-level features reflecting intricate temporal patterns using various dilation rates, which control

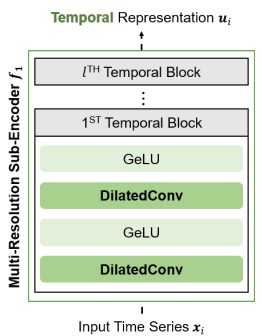

(a) Sub-encoder $f_1$

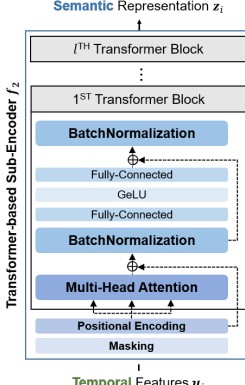

(b) Sub-encoder $f_2$

Figure 3: Configuration of each sub-encoder

how the receptive field of the convolution filter expands across time steps. Moreover, these features enhance the efficiency of the subsequent $f_2$ by leading it to access high-level temporal information obtained from $f_1$. That is, we can achieve the same performance with the relatively shallow $f_2$.

**Transformer-based sub-encoder.** The high-level features $\boldsymbol{u}_i$ are used as inputs for the subsequent sub-encoder $f_2$. As in Figure 3(b), we construct $f_2$ as the transformer introduced in Zerveas et al. (2021). Note that $f_2$ can be flexibly replaced by any model with a similar transformer architecture.

Following the MTM paradigm (Zerveas et al., 2021; Dong et al., 2023), we hide some time steps of $\boldsymbol{u}_i$ with uniformly distributed masks and make the model predict the masked values. In particular, we create a binary mask $\boldsymbol{m}_i \in \mathbb{R}^t$ with a masking ratio $r \in [0.1, 0.9]$. Then, the masked features $\tilde{\boldsymbol{u}}_i \in \mathbb{R}^{t \times d_u}$ are derived by element-wise multiplication: $\tilde{\boldsymbol{u}}_i = \boldsymbol{m}_i \odot \boldsymbol{u}_i$.

---

[1]Here, we show the operation for a single time step for clear presentation; all input values are embedded simultaneously by a single matrix multiplication.

Time series inherently contains information redundancy, enabling the recovery of missing values even with a brief understanding of temporal patterns observed at adjacent time steps. However, when the masking ratio is high, the information redundancy can be eliminated, creating a challenging self-supervisory task that allows the model to identify sophisticated temporal relations. In addition, as shared by most conventional MTMs, exploring an optimal masking ratio for each individual dataset can be time-consuming. Thus, during model training, we randomly pick the masking ratio $r \in [0.1, 0.9]$ for every epoch to avoid searching the optimal masking ratio while enhancing the capability to extract semantic information from $\boldsymbol{u}_i$. In other words, MDTA can capture inherent temporal relations by simultaneously considering a wide range of masking ratios.

Subsequently, since the transformer is insensitive to the ordering of input time series, we add positional encodings $\boldsymbol{\xi}_i \in \mathbb{R}^{t \times d_u}$, obtained by deterministic sinusoidal encoding (Vaswani et al., 2017), to $\tilde{\boldsymbol{u}}_i$ to indicate the sequential nature of the time series: $\tilde{\boldsymbol{u}}_i = \tilde{\boldsymbol{u}}_i + \boldsymbol{\xi}_i$.

Finally, a semantic representation $\boldsymbol{z}_i \in \mathbb{R}^{t \times d_z}$ is generated by

$$\boldsymbol{z}_i = f_2(\boldsymbol{m}_i \odot \boldsymbol{u}_i + \boldsymbol{\xi}_i). \tag{2}$$

**Relation-preserving.** The dual-temporal encoder $f := f_2 \circ f_1$ has a potential risk that temporal information obtained from $f_1$ can be distorted after passing through $f_2$. In other words, as shown in Figures 1(a) and (b), information loss can be caused within $f$ because of its deep sequential architecture, leading to performance degradation by making the representations of some classes indistinguishable. Thus, we introduce a simple yet effective loss function, *relation-preserving*, to maintain temporal structures captured from $f_1$, even after passing $f_2$. To identify structural relations between latent features of $\boldsymbol{u}_i$ along the temporal dimension, we create an adjacency matrix $\mathcal{A}_i \in \mathbb{R}^{d_u \times d_u}$ based on similarities between latent features across time steps of $\boldsymbol{u}_i$ as follows:

$$\mathcal{A}_i = \lfloor \sigma(\boldsymbol{u}_i^\top \boldsymbol{u}_i) \rceil, \tag{3}$$

where $\sigma$ is a sigmoid function that maps input values to the range from zero to one. $\mathcal{A}_i$ is regarded as the ground truth of temporal structures that should be maintained. Then, we obtain another adjacency matrix $\hat{\mathcal{A}}_i \in \mathbb{R}^{d_z \times d_z}$ with $\boldsymbol{z}_i$ generated by $f_2$ as follows:

$$\hat{\mathcal{A}}_i = \sigma(\boldsymbol{z}_i^\top \boldsymbol{z}_i). \tag{4}$$

Note that $d_u$ and $d_z$ should have the same dimension. Finally, we minimize the difference between each element of $\mathcal{A}_i$ and $\hat{\mathcal{A}}_i$ to preserve structural relations from $f_1$ as follows:

$$\mathcal{L}_{RP,i} = - \sum_{a_{pq} \in \mathcal{A}_i, \hat{a}_{pq} \in \hat{\mathcal{A}}_i} a_{pq} \log \hat{a}_{pq} + (1 - a_{pq}) \log(1 - \hat{a}_{pq}), \tag{5}$$

where $a_{pq}$ and $\hat{a}_{pq}$ are the $(p, q)$ element of $\mathcal{A}_i$ and $\hat{\mathcal{A}}_i$, respectively. As shown in Figures 1(c) and (d), this loss function helps to capture intricate temporal patterns in time series effectively and enhances model performance by ensuring a lossless flow of temporal information between $f_1$ and $f_2$. The effect of this loss function is further discussed in Section 4.2.

### 3.2.2 SIMPLE DECODER

Following the MTM paradigm, a decoder only predicts masked values, so its architecture can be flexibly designed regardless of the encoder architecture (He et al., 2022). Thus, we design the lightweight decoder $g$ as one fully connected layer to reduce calculations in the training phase. Note that this decoder reconstructs $\boldsymbol{u}_i$ obtained by $f_1$. In addition, we calculate a mean squared error only for masked content; thereby, *reconstruction* loss function is defined as follows:

$$\mathcal{L}_{RE,i} = \frac{1}{|\mathbb{M}|} \sum_{\tau \in \mathbb{M}} (\hat{\boldsymbol{u}}_{i,\tau} - \boldsymbol{u}_{i,\tau})^2, \tag{6}$$

where $\mathbb{M}$ is a set of indices of masked values, and $\hat{\boldsymbol{u}}_{i,\tau}$ is the reconstructed value by the decoder $g$.

### 3.2.3 CLASSIFICATION HEAD

We employ a classification head $h$ along with $f$ and $g$ to obtain supervisory features, including hard-to-learn class information, from labeled instances. Here, we design $h$ by two fully connected layers with batch normalization and a GeLU activation function.

Given a semantic representation $z_i$ corresponding to $(x_i, y_i) \in \mathbb{D}_\ell$, we first pass $z_i$ through an average pooling layer (AP) and then use it as input for classification head $h$. Formally, we get the predicted class label by $\hat{y}_i = h(\text{AP}(z_i)) = h(\text{AP}(f(x_i)))$, where $f := f_2 \circ f_1$; thereby, *classification* loss function is defined with the cross-entropy as follows:

$$\mathcal{L}_{CL,i} = -y_i \log \hat{y}_i. \tag{7}$$

### 3.2.4 OPTIMIZATION

Following previous works (Xi et al., 2022; Liu et al., 2022), we first train $f$, $g$, and $h$ with the supervised learning using labeled time-series instances and then update $f$ and $g$ with the MTM paradigm using all accessible ones[2]. Specifically, given $\mathbb{D}$ and its subsets $\mathbb{D}_\ell$ and $\mathbb{D}_u$, we train $f$, $g$, and $h$ using $\mathbb{D}_\ell$ by *classification* loss function as follows:

$$\mathcal{L}_\ell = \frac{1}{n_\ell} \sum_{i=1}^{n_\ell} \mathcal{L}_{CL,i}. \tag{8}$$

Subsequently, we update $f$ and $g$ using all accessible instances, including unlabeled ones, in $\mathbb{D} = \mathbb{D}_\ell \cup \mathbb{D}_u$ by *relation-preserving* and *reconstruction* loss functions as follows:

$$\mathcal{L} = \frac{1}{n} \sum_{i=1}^{n} \alpha \mathcal{L}_{RP,i} + \beta \mathcal{L}_{RE,i}, \tag{9}$$

where $\alpha$ and $\beta$ are the weights for each loss. Through this learning process, the labeled instances enable the model to capture useful supervisory features suitable for classification, while all accessible ones enhance the implicit semantic information of time series. The algorithm of the proposed method is summarized in Algorithm 1.

---

**Algorithm 1** Learning procedure of MDTA

---

**Input:** Set of $n$ samples $\mathbb{D}$, its labeled subset $\mathbb{D}_\ell = \{(x_i, y_i)\}_{i=1}^{n_\ell}$ and unlabeled subset $\mathbb{D}_u = \{(x_i)\}_{i=n_\ell+1}^{n}$, dual temporal encoder $f := f_2 \circ f_1$, simple decoder $g$, and classification head $h$
**Output:** Trained $f$, $g$, and $h$
    Initialize $f$, $g$, and $h$.
    **for** each epoch **do**
        **for** $(x_i, y_i) \in \mathbb{D}_\ell$ **do**
            Obtain $u_i$ and $z_i$ by equations (1) and (2), respectively.
            Calculate $\mathcal{L}_{CL,i}$ by equation (7).
        **end for**
        Update $f$, $g$, and $h$ by equation (8).
        **for** $(x_i, \cdot) \in \mathbb{D}$ **do**
            Obtain $u_i$ and $z_i$ by equations (1) and (2), respectively.
            Generate adjacency matrices $\mathcal{A}_i$ and $\hat{\mathcal{A}}_i$ by equations (3) and (4), respectively.
            Calculate $\mathcal{L}_{RP,i}$ and $\mathcal{L}_{RE,i}$ by equations (5) and (6), respectively.
        **end for**
        Update $f$, $g$, and $h$ by equation (9).
    **end for**

---

## 4 EXPERIMENTAL RESULTS

We evaluated the model performance of the proposed method, MDTA, on semi-supervised time-series classification compared to seven baselines on 15 univariate time-series classification datasets from the UCR archive (Dau et al., 2018). Here, we describe the experimental results in detail. The experimental setting, such as datasets and baseline methods, and implementation details are provided in Sections A and B, respectively.

---

[2]To enrich the inherent semantic information of time series, we also use labeled time series as well as unlabeled ones to train $f$ and $g$ following the MTM paradigm.

Table 1: Average classification performance across label ratios ranging from 0.1 to 0.9 under *inductive* inference for MDTA and baselines. For each dataset, the best score is highlighted in boldface.

| Dataset | CE | Pseudo | Π-model | FixMatch | MTL | SSTSC | iTimes | MDTA (ours) |
|---|---|---|---|---|---|---|---|---|
| CBF | 99.20 | 99.30 | 99.26 | 99.40 | 98.38 | 99.38 | 99.44 | **99.71** |
| CricketX | 52.26 | 52.62 | 61.92 | 59.66 | 40.38 | **67.04** | 64.07 | |
| ECGFiveDays | 98.49 | 98.51 | 83.44 | 83.33 | 98.39 | 98.24 | 95.33 | **99.81** |
| Lightning2 | 67.56 | 68.89 | 68.98 | 69.87 | 67.56 | 67.64 | 71.56 | **75.23** |
| MoteStrain | 93.29 | 93.46 | 94.15 | 94.19 | 89.17 | 92.31 | 93.56 | **95.03** |
| Plane | 96.98 | **96.98** | 85.98 | 88.73 | 84.87 | 94.50 | 85.66 | 96.88 |
| PowerCons | 89.41 | 88.89 | 85.37 | 86.23 | 87.84 | 89.07 | 87.78 | **93.58** |
| RefrigerationDevices | 58.60 | 57.61 | 57.04 | 57.32 | 57.51 | 58.19 | **60.64** | 59.56 |
| SonyAIBORobotSurface1 | 97.35 | 97.21 | 93.44 | 94.22 | 93.28 | 96.80 | 94.20 | **99.61** |
| SwedishLeaf | 84.16 | 84.83 | 70.34 | 69.66 | 55.45 | 76.93 | 56.18 | **86.18** |
| SyntheticControl | 96.19 | 97.70 | 97.98 | 97.52 | 96.98 | 93.78 | 97.31 | **98.11** |
| ToeSegmentation1 | 84.16 | 84.44 | 84.73 | 84.36 | 82.30 | 82.39 | 86.71 | **94.03** |
| Trace | 91.94 | 93.22 | 91.72 | 92.72 | 91.50 | 91.44 | 95.78 | **98.44** |
| TwoPatterns | 99.81 | 99.85 | 99.30 | 99.48 | 98.91 | 99.73 | 96.86 | **99.97** |
| Yoga | 83.99 | 83.98 | 64.72 | 63.85 | 74.76 | 80.58 | 75.92 | **85.17** |
| *Average Rank* | 4.27* | 3.67* | 5.20* | 5.07* | 6.93* | 5.27* | 4.27* | **1.27** |
| *(p-value)* | (1.22e-4) | (1.22e-4) | (6.10e-5) | (6.10e-5) | (6.10e-5) | (6.10e-5) | (2.01e-3) | - |

## 4.1 SEMI-SUPERVISED TIME-SERIES CLASSIFICATION

The model performance on semi-supervised time-series classification can be evaluated by inductive and transductive inferences. The inductive inference involves measuring model performance using a test dataset separate from the training dataset, while the transductive inference evaluates model performance for unlabeled instances in the training dataset. Here, referring to Xi et al. (2022) and Liu et al. (2022), we focus on providing the results under inductive inference, but those under transductive inference are also given in Table 8 in Section J.

Table 1 depicts the classification performance of MDTA compared to the baselines for 15 time-series datasets. Here, we show the classification performance by averaging accuracy scores across label ratios from 0.1 to 0.9 on each dataset. The complete results with standard deviations for different label ratios on each dataset are provided in Table 7 in Section J. Moreover, we performed statistical tests on the classification performance to ensure the significance of performance improvement by MDTA. Specifically, we employed a two-sample Wilcoxon signed rank test between MDTA and each baseline. The superscript * for the average rank in Table 1 implies the rank test's p-value was smaller than 0.01.

MDTA achieved the best performance in 12 out of 15 datasets, and the average rank was 1.27, remarkably outperforming the baselines. Moreover, the proposed method showed relatively lower standard deviations than the baselines in most datasets (see Table 7 in Section J), and the statistical test shows that MDTA performed significantly better than the baselines. When the labels were highly limited, MDTA also exhibited outstanding performance in most datasets (see Table 7 in Section J). In particular, as in Figure 4, MDTA improved classification

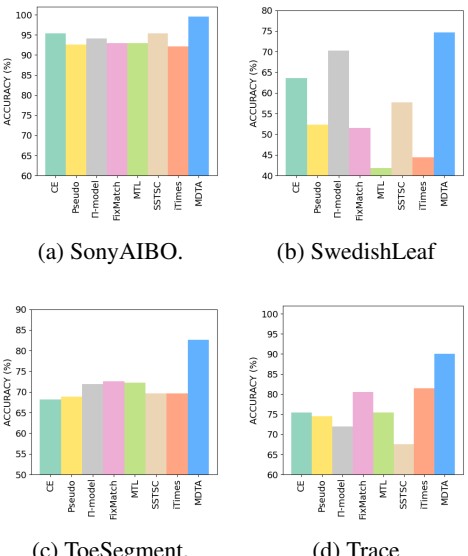

(a) SonyAIBO.   (b) SwedishLeaf

(c) ToeSegment.   (d) Trace

Figure 4: Accuracy scores of MDTA and baselines on (a) *SonyAIBORobotSurface1*, (b) *SwedishLeaf*, (c) *ToeSegmentation1*, and (d) *Trace*, when label ratio is 0.1.

performance by more than 4% in four datasets, *SonyAIBORobotSurface1*, *SwedishLeaf*, *ToeSegmentation1*, and *Trace*, than the second-best scores when the label ratio was 0.1. These results support the effectiveness of MDTA in leveraging unlabeled instances.

In addition, our method showed superior performance in most datasets regardless of the number of classes and sequence length by effectively incorporating intrinsic semantic information of time

series with supervisory features. By contrast, the baselines exhibited performance differences according to the characteristics of time-series datasets. For example, *Pseudo*, the second-best method in average rank, showed relatively low performance in the datasets with long sequences, such as *RefrigerationDevices* and *Lightning2* datasets, because it does not consider temporal dependency of time series; *iTimes*, one of the SOTAs in semi-supervised time-series classification, performed poorly in *SwedishLeaf*, the dataset with the largest number of classes, and *SonyAIBORobotSurface1*, that with the shortest average sequence length.

## 4.2 ABLATION STUDIES

MDTA has three key components: 1) **dual-temporal encoder** architecture that effectively captures intricate temporal structures with diverse resolutions by two sequential sub-encoders, 2) **relation-preserving** loss function that prevents information loss within the encoder, and 3) **random masking ratios** to avoid the effort of exploring optimal masking ratios while enhancing model performance.

To demonstrate their effectiveness, we compared MDTA to three ablation models: MDTA with that the multi-resolution sub-encoder in the dual-temporal encoder is replaced by one fully connected layer (*MDTA w/o $f_1$*), MDTA without relation-preserving (*MDTA w/o $\mathcal{L}_{RP}$*), and MDTA without random masking ratios (*MDTA w/o RM*). For *MDTA w/o RM*, we fixed the masking ratio to 0.5. The classification performance of ablation models and MDTA is listed in Table 2. Here, we show the results by averaging accuracy scores across label ratios from 0.1 to 0.9 on each dataset. The complete results for all label ratios on each dataset are given in Table 9 in Section J.

Table 2: Average classification performance across label ratios ranging from 0.1 to 0.9 for MDTA and the ablation models. For each dataset, the best score is highlighted in boldface.

| Dataset | MDTA (ours) | MDTA w/o $f_1$ | MDTA w/o $\mathcal{L}_{RP}$ | MDTA w/o RM |
|---|---|---|---|---|
| CBF | **99.71** | 99.60 | 99.08 | 99.22 |
| CricketX | **64.07** | 34.24 | 58.78 | 61.32 |
| ECGFiveDays | 99.81 | 82.49 | **99.87** | 99.56 |
| Lightning2 | **75.23** | 67.85 | 72.59 | 67.41 |
| MoteStrain | **95.03** | 90.38 | 93.03 | 94.52 |
| Plane | 96.88 | 93.83 | **97.27** | 96.12 |
| PowerCons | **93.58** | 86.78 | 92.70 | 92.23 |
| RefrigerationDevices | 59.56 | 53.85 | **60.07** | 58.22 |
| SonyAIBORobotSurface1 | **99.61** | 93.75 | 99.38 | 99.35 |
| SwedishLeaf | **86.18** | 68.89 | 83.60 | 83.36 |
| SyntheticControl | **98.11** | 83.86 | 96.85 | 97.99 |
| ToeSegmentation1 | **94.03** | 90.74 | 91.98 | 91.91 |
| Trace | 98.44 | 94.54 | **98.52** | 95.74 |
| TwoPatterns | **99.97** | 98.11 | 99.88 | 99.93 |
| Yoga | **85.17** | 77.33 | 78.29 | 84.16 |
| *Average Decline Rate* (%) | - | 9.60 | 1.75 | 1.81 |

**Dual-temporal encoder.** In general, temporal dependencies, one of the unique characteristics of time series, span various time intervals within a time series; hence, reflecting diverse resolutions enables the model to recognize the temporal dependencies easily, improving classification performance (Zerveas et al., 2021). Thus, we designed the *dual-temporal encoder $f$* with the multi-resolution sub-encoder $f_1$ to allow the model to capture intrinsic temporal patterns at different time scales. To examine its effect, we compared the classification performance of MDTA and *MDTA w/o $f_1$*. As shown in Table 2, the performance of *MDTA w/o $f_1$* highly decreased by approximately 9.60% compared to that of MDTA on average. Thus, we demonstrated that the proposed encoder architecture effectively improves classification performance by considering diverse temporal resolutions.

**Relation-preserving.** The proposed encoder architecture comprising two sequential sub-encoders has one potential risk: temporal information obtained from the multi-resolution sub-encoder, $f_1$, can be corrupted by passing through the transformer-based sub-encoder, $f_2$. To address this risk, we introduced *relation-preserving, $\mathcal{L}_{RP}$*, that minimizes the temporal structural difference between representations generated by $f_1$ and $f_2$. We conducted two experiments to investigate the effects of this loss function on preventing temporal information loss and improving classification performance.

The information loss can be more severe when the number of layers in $f_2$ is larger. We constructed four models, each with either 5 or 20 layers in $f_2$, and either the relation-preserving loss term or

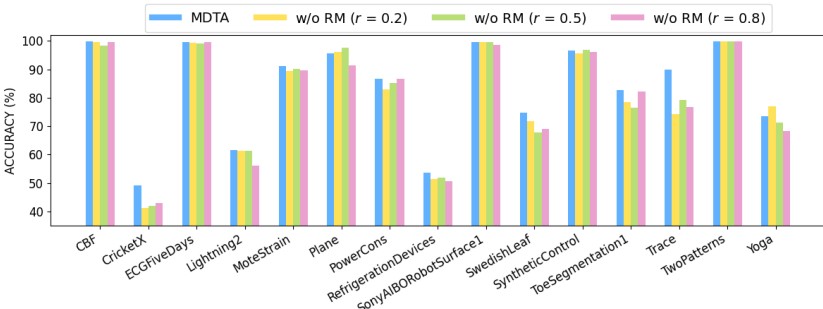

Figure 5: Accuracy scores of MDTA and the ablation models using fixed masking ratios, $r$, of 0.2, 0.5, and 0.8, respectively. Here, we provide the scores when the label ratio is 0.1 because it best demonstrates the effect of masking ratios in capturing semantic information of time series.

not. Here, we trained those models with 100 epochs when the label ratio was 0.1. As in Figures 1(a) and (b), the models without relation-preserving caused information loss between two sub-encoders by mixing the representations of some classes when the number of layers is large. In this case, the model with 20 layers in $f_2$ achieved an accuracy of 70%. By contrast, in Figures 1(c) and (d), the models with relation-preserving learned the representations to successfully discriminate for all classes with an accuracy of 96.67%, even after passing through $f_2$ with 20 layers. We further provide the graphical analysis of MDTA in forming distinct groups for each class in Figure 7 in Section F.

Also, as in Table 2, *MDTA w/o $\mathcal{L}_{RP}$* exhibited 1.75% lower classification performance than MDTA on average. Thus, we demonstrated that the relation-preserving ensures a lossless flow of temporal information between two sub-encoders, enhancing classification performance.

**Random masking ratios.** One practical drawback of conventional MTMs is their sensitivity to masking ratios. As shown in Figure 5, when we fix the masking ratio to a certain value, the performances highly vary by the datasets. However, searching for an optimal masking ratio for each dataset is often impractical. Thus, we used *random masking ratios* to avoid exploring optimal masking ratios while enhancing the capability to capture temporal relations within time series. Consequently, Figure 5 and Table 2 demonstrate that random masking ratios enhance the model's generalization performance without the high-cost tuning process for finding optimal ratios (see Section G.1).

Furthermore, the random masking ratios help the model to identify intricate temporal relations; thereby, the model can be robust to missing values occurring in the inference phase because the model can easily recover the missing parts. We validated the better robustness against missing values of the proposed method than the models with the fixed masking ratios (see Section G.2).

## 5 CONCLUSION

We proposed a novel MTM-based framework, MDTA, for semi-supervised time-series classification. MDTA effectively captures semantic information of time series by reflecting diverse temporal resolutions without information loss within the dual-temporal encoder and using random masking ratios. Then, we incorporated the extracted semantic information from unlabeled instances with supervisory features obtained from labeled ones to enhance classification performance. Through extensive experiments on semi-supervised time-series classification, we demonstrated that MDTA is effective for capturing semantic information on time series as well as performs better than SOTAs.

Nevertheless, the proposed method is relatively inefficient compared to the other baselines because it employs two consecutive sub-encoders (see Table 6). Although we used the transformer architecture introduced in Zerveas et al. (2021), the first proposed transformer for time series to extract their useful representations, its computational inefficiency has also been demonstrated in Cheng et al. (2023). In other words, the efficiency of our approach can be improved by replacing the transformer architecture with one of the recent transformers, improving computational efficiency. In addition, another possible solution for future research direction is to devise a lightweight transformer encoder architecture that can reflect diverse temporal resolutions.

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

# A    EXPERIMENTAL SETTINGS

**Baselines.**    To demonstrate the efficacy of the proposed method, we compared MDTA with seven baseline methods, including SOTAs in semi-supervised time-series classification and a fully supervised model as follows:

- *CE* is trained in a supervised manner using cross-entropy loss for only labeled instances.
- *Pseudo* (Lee et al., 2013) is a semi-supervised learning approach that generates pseudo-labels of unlabeled instances based on the current model prediction to supervise them.
- Π-*model* (Laine & Aila, 2016) is a semi-supervised method that incorporates consistency regularization into pseudo-labeling, exploiting relationships between labeled and unlabeled instances during training.
- *FixMatch* (Sohn et al., 2020) is a semi-supervised learning method that creates high-confident pseudo-labels for weakly augmented unlabeled instances and uses these labels to supervise strongly augmented instances.
- *MTL* (Jawed et al., 2020) is a semi-supervised time-series classification method combining self-supervised learning and multi-task learning.
- *SSTSC* (Xi et al., 2022) is a SOTA in semi-supervised time-series classification that considers the inherent temporal information of time series by exploring the temporal relations between past, present, and future.
- *iTimes* (Liu et al., 2022) is another SOTA in semi-supervised time-series classification that captures the temporal structure of unlabeled instances by training the model to recognize time sampling functions that are randomly applied to input time series.

**Datasets.**    We used 15 univariate time-series classification datasets from the UCR time-series classification archive (Dau et al., 2018). Due to the limited computing resources and time, it was difficult to utilize all datasets in the UCR archive. Hence, as many previous studies did (Jawed et al., 2020; Fan et al., 2021; Xi et al., 2022; Liu et al., 2022; Eldele et al., 2023), we selected some datasets with various data types, quantities, and sequence lengths. A detailed description of the datasets is listed in Table 3. The archive separately provided training and test datasets; hence, we used the training dataset to train the model and the test dataset to evaluate the trained model. In addition, we set 20% of the training data as a validation dataset and split the remaining training dataset into labeled and unlabeled instances according to label ratios. All instances were normalized with a z-score. Note that the proposed method can also cover multivariate time series, although we confined the evaluation to univariate time series ($v = 1$) to ensure fair comparisons with the baselines (Xi et al., 2022; Liu et al., 2022).

Table 3: Detailed description of 15 datasets. The data type, numbers of data and classes, and sequence lengths are provided for each dataset.

| Dataset | Abbreviation | Data Type | Number of Data | Number of Classes | Sequence Length |
|---|---|---|---|---|---|
| CBF | CB | Simulated | 930 | 3 | 128 |
| CricketX | CX | Motion | 780 | 12 | 300 |
| ECGFiveDays | EF | ECG | 884 | 2 | 136 |
| Lightning2 | L2 | Sensor | 121 | 2 | 637 |
| MoteStrain | MS | Sensor | 1272 | 2 | 84 |
| Plane | PL | Sensor | 210 | 7 | 144 |
| PowerCons | PC | Power | 360 | 2 | 144 |
| RefrigerationDevices | RD | Device | 750 | 2 | 720 |
| SonyAIBORobotSurface1 | SR | Sensor | 621 | 2 | 70 |
| SwedishLeaf | SL | Image | 1125 | 15 | 128 |
| SyntheticControl | SC | Simulated | 600 | 6 | 60 |
| ToeSegmentation1 | T1 | Motion | 268 | 2 | 277 |
| Trace | TR | Sensor | 200 | 4 | 275 |
| TwoPatterns | TP | Simulated | 5000 | 4 | 128 |
| Yoga | YO | Image | 3300 | 2 | 426 |

**Evaluation metric.**    We evaluated the classification performance by measuring the accuracy score.

## B  IMPLEMENTATION DETAILS

Time-series datasets have been collected from various sources, and their label ratio also varies; hence, selecting appropriate hyperparameters for each dataset is impractical. Therefore, we used fixed hyperparameters regardless of the datasets and their label ratios to avoid impractical tuning efforts (Yue et al., 2022). In MDTA, the multi-resolution sub-encoder $f_1$ contained four temporal blocks, each comprising two *DilatedConv*s with GeLU activation functions (see Figure 3(a)), and the skip connections were introduced between neighboring blocks. For the $l$-th block, the dilation rate $\rho$ was set as $2^l$. The kernel size was set to 3; each *DilatedConv* had a dimension of 16; a residual block mapped the hidden features to $d_u$-dimensional temporal features, $\boldsymbol{u}_i$. Subsequently, the sub-encoder $f_2$ consisted of the transformer block with batch normalization proposed in Zerveas et al. (2021). In particular, we configured $f_2$ with eight heads for multi-head attention and three transformer blocks (see Figure 3(b)). The dimensions of two fully connected layers in each transformer block were set to 256 and $d_z$. Then, $f_2$ mapped the temporal features $\boldsymbol{u}_i$ to a $d_z$-dimensional representation $\boldsymbol{z}_i$. The decoder $g$ was designed as a fully connected layer, reconstructing the semantic representation $\boldsymbol{z}_i$ generated by $f_2$ into a $d_u$-dimensional temporal features $\hat{\boldsymbol{u}}_i$. The classification head $h$ was constructed with two fully connected layers with batch normalization and a GeLU activation function, with a hidden dimension 256. This head produced predicted class labels $\hat{y}_i$ using $\boldsymbol{z}_i$ of $\boldsymbol{x}_i$ obtained by $f := f_2 \circ f_1$ as input. The dimensions $d_u$ and $d_z$ were both set to 64. Moreover, the loss weights, $\alpha$ and $\beta$, used in equation (9) for the MTM paradigm are set differently at every epoch by the adaptive loss weighting strategy introduced in Heydari et al. (2019).

Table 4: Average classification performance across label ratios ranging from 0.1 to 0.9 for the baselines with different encoder architectures. The higher average score is highlighted in boldface for each baseline. ($\mathbb{D}$: Dataset, *Trans*: transformer architecture)

| $\mathbb{D}$ | CE | | Pseudo | | Pi | | FixMatch | | MTL | | SSTSC | | iTimes | |
|---|---|---|---|---|---|---|---|---|---|---|---|---|---|---|
| | *Trans* | SimConv | *Trans* | SimConv | *Trans* | SimConv | *Trans* | SimConv | *Trans* | SimConv | *Trans* | SimConv | *Trans* | SimConv |
| CB | 99.61 | 99.20 | 99.61 | 99.30 | 99.45 | 99.26 | 93.20 | 99.40 | 98.63 | 98.38 | 99.71 | 99.38 | 99.67 | 99.44 |
| CX | 31.44 | 52.26 | 31.38 | 52.62 | 62.17 | 61.92 | 26.60 | 59.66 | 24.29 | 40.38 | 31.67 | 41.89 | 36.38 | 67.04 |
| EF | 76.22 | 98.49 | 77.07 | 98.51 | 84.77 | 83.44 | 74.12 | 83.33 | 74.50 | 98.39 | 76.20 | 98.24 | 77.94 | 95.33 |
| L2 | 60.89 | 67.56 | 63.38 | 68.89 | 68.18 | 68.98 | 59.11 | 69.87 | 60.71 | 67.56 | 67.02 | 67.64 | 75.20 | 71.56 |
| MS | 87.50 | 93.29 | 88.83 | 93.46 | 94.20 | 94.15 | 85.53 | 94.19 | 87.28 | 89.17 | 89.89 | 92.31 | 91.90 | 93.56 |
| PL | 89.10 | 96.98 | 87.88 | 96.98 | 85.45 | 85.98 | 48.57 | 88.73 | 69.52 | 84.87 | 75.03 | 94.50 | 52.22 | 85.66 |
| PC | 86.73 | 89.41 | 86.36 | 88.89 | 86.11 | 85.37 | 84.44 | 86.23 | 86.30 | 87.84 | 85.59 | 89.07 | 85.31 | 87.78 |
| RD | 61.69 | 58.60 | 62.46 | 57.61 | 57.13 | 57.04 | 58.99 | 57.32 | 61.87 | 57.51 | 62.50 | 58.19 | 60.21 | 60.64 |
| SR | 91.57 | 97.35 | 90.99 | 97.21 | 94.22 | 93.44 | 89.21 | 94.22 | 82.20 | 93.28 | 91.18 | 96.80 | 92.50 | 94.20 |
| SL | 60.38 | 84.16 | 62.51 | 84.83 | 71.92 | 70.34 | 22.82 | 69.66 | 31.86 | 55.45 | 57.91 | 76.93 | 52.55 | 56.18 |
| SC | 91.00 | 96.19 | 91.59 | 97.70 | 97.24 | 97.98 | 69.67 | 97.52 | 80.15 | 96.98 | 85.93 | 93.78 | 89.07 | 97.31 |
| T1 | 78.56 | 84.16 | 78.48 | 84.44 | 85.10 | 84.73 | 76.34 | 84.36 | 72.80 | 82.30 | 76.50 | 82.39 | 78.19 | 86.71 |
| TR | 98.78 | 91.94 | 94.00 | 93.22 | 94.39 | 91.72 | 98.22 | 92.72 | 91.78 | 91.50 | 97.39 | 91.44 | 99.61 | 95.78 |
| TP | 51.64 | 99.81 | 61.02 | 99.85 | 99.34 | 99.30 | 28.41 | 99.48 | 83.52 | 98.91 | 89.18 | 99.73 | 72.68 | 96.86 |
| YO | 64.13 | 83.99 | 65.23 | 83.98 | 66.56 | 64.72 | 58.00 | 63.85 | 66.30 | 74.76 | 77.90 | 80.58 | 64.05 | 75.92 |
| *Average* | 75.28 | **86.23** | 76.05 | **86.50** | 83.08 | 82.56 | 64.88 | **82.70** | 71.45 | **81.15** | 77.57 | **84.19** | 75.17 | **84.26** |

Following Xi et al. (2022) and Liu et al. (2022), for the baseline methods, we employed a simple four-layer convolutional neural network (SimConv) with ReLU activation function and batch normalization as a backbone encoder. In particular, the dimensions of four layers were set to 8, 16, 32, and 64, respectively; the kernel size was set to 4, and the stride to 2 for every layer. The encoder architecture of baselines differed from MDTA due to the better performance of SimConv compared to the transformer architecture in most baselines. Table 4 shows the classification performance of the baseline methods by averaging accuracy scores across label ratios from 0.1 to 0.9 on each dataset. The classifier for the baselines was configured with the same architecture as the classification head $h$ of the proposed method. We used time-warping and magnitude-warping augmentations for all baselines during model training (Xi et al., 2022; Liu et al., 2022). The baseline methods, except iTimes, were implemented based on the official code of SSTSC[3]; iTimes was implemented with the code provided by its authors[4].

We set the batch size and maximum training epoch to 10 and 1000, respectively, for all methods, including the proposed one. We used the Adam optimizer with weight decay (AdamW) (Loshchilov & Hutter, 2017) with a learning rate of 0.001 for model training. Also, we adapted an early stopping strategy with a patience of 50 epochs based on validation accuracy to efficient model training. We repeated the experiments five times and reported the average and standard deviation.

---

[3]https://github.com/mrxiliang/sstsc
[4]1909030127@stu.hrbust.edu.cn

All experiments were executed using the Pytorch platform on a system with an Intel Core i9-10900X CPU clocked at 3.70 GHz, 256 GB RAM, and GeForce RTX 3090 24GB GPU. The source code is attached as a zip file in the submission.

## C   COMPARISON WITH CA-TCC

The proposed method can be distinguished from CA-TCC (Eldele et al., 2023) in the architectures and purposes of using convolutional neural network (CNN) and transformer. First, CA-TCC used the Residual network introduced in Wang et al. (2017) as a CNN encoder to extract high-level representations to be augmented for contrastive learning because contrastive learning in the latent space generally performs better than that in the data space (Yue et al., 2022). By contrast, the proposed method used CNN with causal padding and dilated filters as the former sub-encoder to reflect various temporal resolutions while enhancing the efficiency of the latter transformer-based sub-encoder (Fan et al., 2020).

Second, CA-TCC just employed the transformer to extract useful representations used for contextual contrasting. In contrast, the proposed method used the transformer as the encoder architecture of the masked autoencoder for masked time-series modeling, which is effective in capturing fine-grained semantic information of time series.

To summarize, CA-TCC is a contrastive learning-based semi-supervised time-series classification method, which inherits the limitations of the existing contrastive learning methods and transformer architectures: the high sensitivity to data augmentations and the impossibility of considering diverse temporal resolutions. By contrast, we proposed a masked time-series modeling-based semi-supervised time-series classification framework by considering diverse temporal resolutions and random masking ratios. In addition, we did not use contrastive learning, so any data augmentations with strong inductive biases are not required (Yue et al., 2022).

Furthermore, we compared the proposed method with the most recent work, CA-TCC. Table 5 shows the average accuracy scores across label ratios from 0.1 to 0.9 under inductive inference. The proposed method performs better than CA-TCC on average, achieving better performance in 10 out of 15 datasets. Especially for several datasets, such as *PowerCons*, *SonyAIBORobotSurface1*, *Swedish-Leaf*, *SyntheticControl*, *ToeSegmentation1*, *Trace*, and *Yoga*, our method shows overwhelming performance compared to CA-TCC. In contrast, the performance gaps in five datasets that CA-TCC beats our method are relatively small.

Table 5: Average classification performance across label ratios ranging from 0.1 to 0.9 for MDTA and CA-TCC. The higher average score is highlighted in boldface for each baseline.

| Method | Dataset | | | | | | | | | | | | | | |
|---|---|---|---|---|---|---|---|---|---|---|---|---|---|---|---|
| | CB | CX | EF | L2 | MS | PL | PC | RD | SR | SL | SC | T1 | TR | TP | YO |
| CA-TCC | **99.84** | **71.03** | 97.27 | **75.67** | 93.24 | 96.35 | 87.20 | **62.26** | 77.90 | 77.67 | 91.85 | 67.87 | 81.94 | **100.00** | 80.76 |
| MDTA | 99.71 | 64.07 | **99.81** | 75.23 | **95.03** | **96.88** | **93.58** | 59.56 | **99.61** | **86.18** | **98.11** | **94.03** | **98.44** | 99.97 | **85.17** |

## D   COMPARISON WITH SEMITIME

The proposed method and SemiTime (Fan et al., 2021) are totally different regarding the aims and approaches, although they share the binary cross-entropy loss.

Our relation-preserving loss function aims to prevent the loss of the temporal structural information obtained from the former sub-encoder while passing through the subsequent sub-encoder, whereas the binary cross-entropy used in SemiTime aims to capture temporal dependency by using the past and future segments. Therefore, our relation-preserving loss function minimizes the difference between gram matrices derived from the outputs of two sub-encoders, whereas SemiTime encourages the positive pairs of segments to be consistent and the negative pairs to be distant.

## E   VISUALIZATION OF RELATION-PRESERVING

Here, we present Figure 6 that is enlarged Figure 1 to enhance visibility.

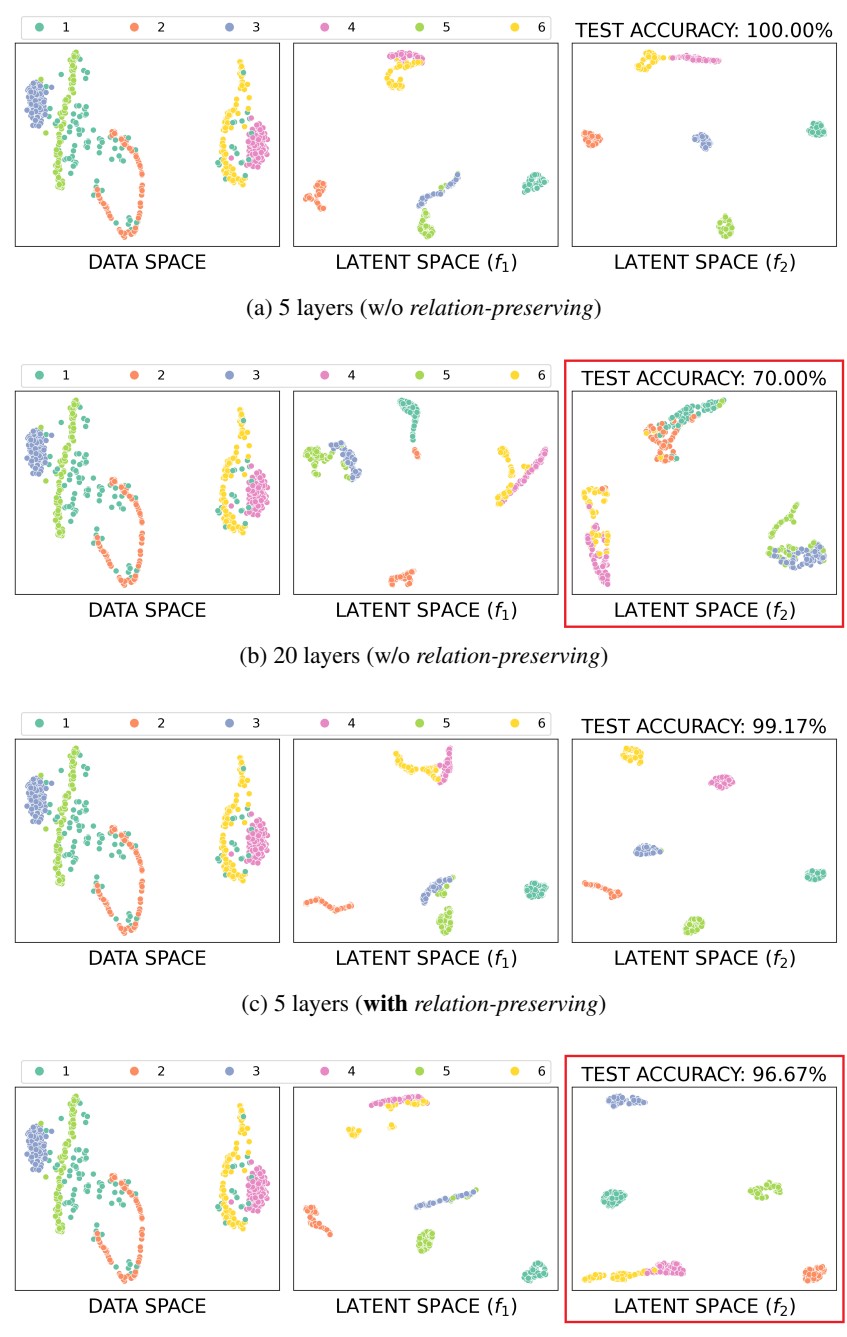

Figure 6: The enlarged Figure 1 —Visualization of data space and latent spaces produced by the sequential sub-encoders, $f_1$ and $f_2$, for *SyntheticControl* dataset. In (a) and (b), we present each space when $f_2$ has 5 and 20 layers, respectively, without *relation-preserving*. In contrast, (c) and (d) show each space when $f_2$ has 5 and 20 layers, respectively, with *relation-preserving*. In (b), representations of each class obtained from $f_2$ exhibit diminished distinctiveness compared to those generated by $f_1$. However, in (d), representations of each class remain distinguishable.

## F GRAPHICAL ANALYSIS

As in Figure 7, we performed a graphical analysis for the representations learned by MDTA using UMAP (McInnes et al., 2018). We selected the six largest datasets: *CBF*, *ECGFiveDays*, *MoteS-train*, *SwedishLeaf*, *TwoPatterns*, and *Yoga*. For each dataset, we compared the data space and latent spaces produced by the sequential sub-encoders, $f_1$ and $f_2$, using all accessible instances. Consequently, we observed that as the representations of each class pass through each sub-encoder, they form gradually more distinct groups for each class. Through this analysis, we can reaffirm the effectiveness of the proposed method, especially for the dual-temporal encoder architecture and relation-preserving loss function.

## G EFFECTS OF RANDOM MASKING RATIOS

### G.1 GENERALIZATION PERFORMANCE

In general, exploring optimal masking ratios for each individual dataset within reasonable time and cost is impractical. In addition, if we consider various masking ratios during model training, the information redundancy originating from the correlation between time steps can be eliminated from diverse perspectives; thereby, a variety of challenging self-supervisory tasks that allow the model to identify sophisticated temporal relations are created. Therefore, we used random masking ratios to enhance model generalization performance by identifying intricate temporal relations without the inefficiency of searching for proper masking ratios.

To demonstrate this effect of random masking ratios, in Figure 5, we compared the random masking ratio with three fixed masking ratios of 0.2 (low), 0.5 (medium), and 0.8 (high). Although these fixed masking ratios may not be optimal for every dataset, we can examine the overall tendency of each dataset against the low, medium, and high masking ratios. Consequently, when we fix the masking ratio to a certain value, the performances highly vary by the datasets. In contrast, the random masking ratio achieved the best performance in 12 out of 15 datasets and also showed decent performances in the remaining datasets.

In addition, in Table 2, we compared the average classification performance of the random masking ratio with that of the fixed ratio of 0.5 in more detail. Consequently, the average classification performance of the fixed ratio of 0.5 showed a drop rate of 1.81% compared to the proposed random masking ratio. Specifically, as shown in Table 9, which provides the complete results, the performance of the fixed masking ratio of 0.5 remarkably decreased by over 10% in several cases, especially for low label ratios on some datasets. Therefore, we demonstrate the random masking ratio enhances the generalization performance of the model without the high-cost tuning process for finding optimal masking ratios.

### G.2 ROBUSTNESS TO MISSING VALUES

By masking several time steps and predicting the masked parts, inherent temporal relations that are not easily identified can be captured. Then, the model can be robust to missing values because it can relatively easily infer the missing parts. Thus, regarding robustness against missing values, the MTM paradigm can prevent a drastic performance decrease through the captured temporal structures, even if some missing values occur in the inference phase.

MDTA enhances the capability to capture complex temporal relations within time series using random masking ratios during model training. Thus, our method is more robust against missing values than MDTAs with fixed masking ratios. To confirm this, we analyzed the robustness against missing values of MDTA. We compared the classification performance of MDTA with those of the models with the fixed masking ratios $r$ of 0.2, 0.5, and 0.8, respectively, when the missing ratio varies from 0.1 to 0.9 for input time series in the inference phase. Here, we set the label ratio to 0.5.

Consequently, in Figure 8, the proposed method was more robust to missing values than the models with the fixed masking ratios in most datasets by considering a wide range of masking ratios in the training phase. Especially in *Trace*, MDTA maintained decent performance even with high missing ratios. However, when it is difficult to capture temporal structures, e.g., missing ratios are over 0.5, the model performance is likely to decrease even using random masking ratios.

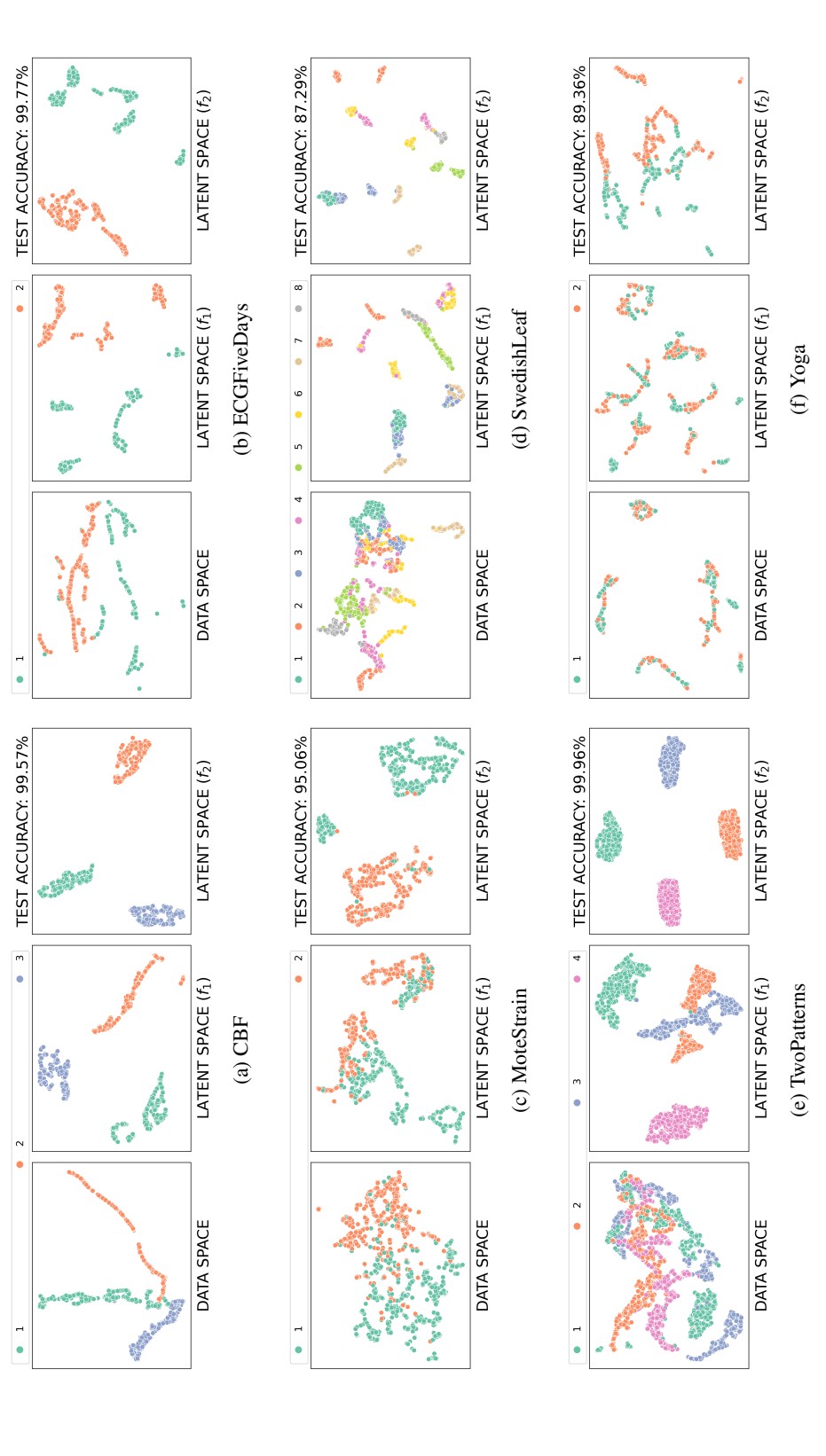

Figure 7: Visualization of data space and latent spaces from $f_1$ and $f_2$ for the six largest datasets: (a) *CBF*, (b) *ECGFiveDays*, (c) *MoteStrain*, (d) *SwedishLeaf*, (e) *TwoPatterns*, and (f) *Yoga*

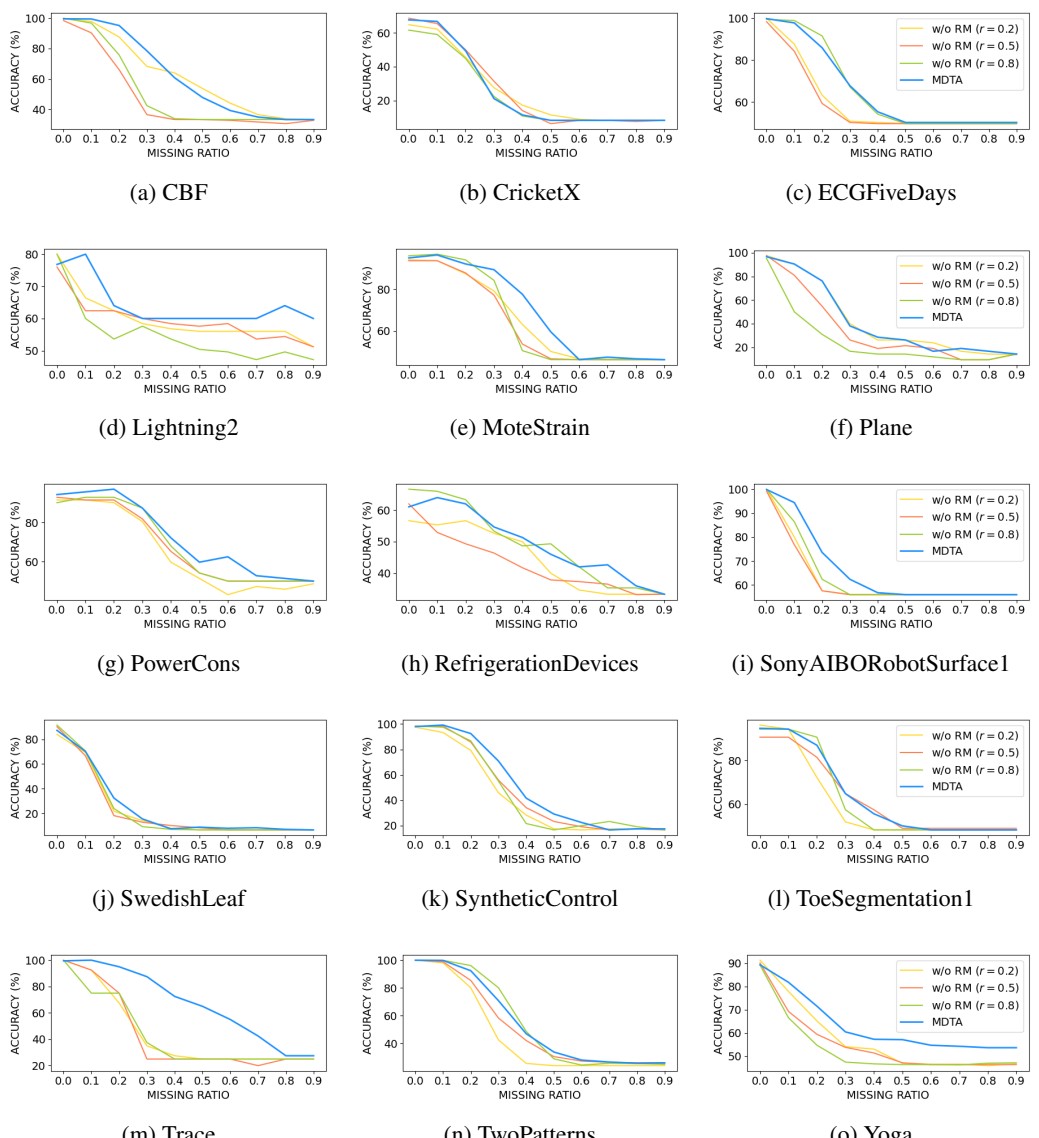

Figure 8: Accuracy scores of MDTA and the ablation models using fixed masking ratios of 0.2, 0.5, and 0.8, respectively, when varying missing ratios $\in [0.1, 0.9]$

# H  SENSITIVITY ANALYSIS ON LOSS WEIGHTS

To examine the impact of loss weights, $\alpha$ and $\beta$, used in equation (9) for the MTM paradigm, we performed sensitivity analysis against them. We compared the adaptive loss weighting strategy (Heydari et al., 2019) to three fixed loss weights. Here, the pairs of $\alpha$ and $\beta$ for the fixed loss weights were set to (0.2, 0.8), (0.5, 0.5), and (0.8, 0.2). Consequently, as shown in Figure 9, the adaptive loss weighting exhibits comparable performance with the three fixed loss weights on average. However, in some datasets, such as *CricketX*, *SwedishLeaf*, and *Yoga*, the adaptive loss weighting performs notably better than the others. In addition, it allows us to reduce the effort for finding optimal values for $\alpha$ and $\beta$ on each dataset. Therefore, we used the adaptive loss weighting strategy for the experiments performed in this paper.

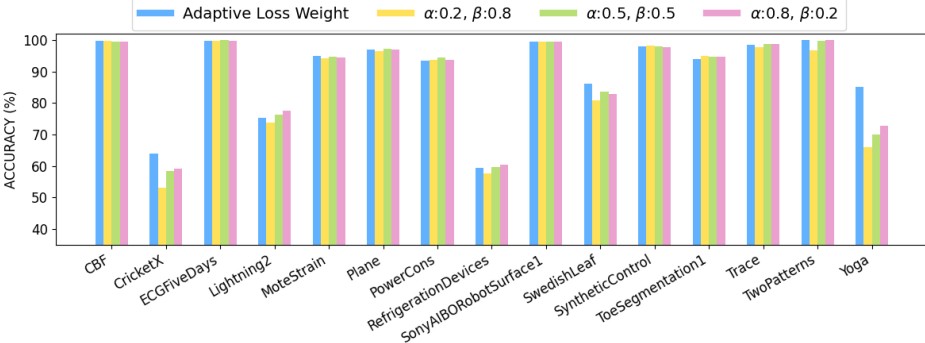

Figure 9: Accuracy scores for different $\alpha$ and $\beta$. Here, we compared the adaptive loss weighting strategy and three fixed loss weights.

## I TRAINING EFFICIENCY

Table 6 shows the average training time per epoch of the proposed method with that of baseline methods across label ratios from 0.1 to 0.9 on 15 datasets. As mentioned in 5, MDTA performs slower than other baselines because it employs two sequential consecutive sub-encoders.

Table 6: Average training time (sec) per epoch of the baselines and MDTA across label ratios from 0.1 to 0.9 on each dataset

| Dataset | Pseudo | $\Pi$-model | FixMatch | MTL | SSTSC | iTimes | MDTA |
|---|---|---|---|---|---|---|---|
| CBF | 0.57 | 0.97 | 0.98 | 0.90 | 1.39 | 0.93 | 5.74 |
| CricketX | 0.48 | 0.83 | 0.85 | 0.79 | 1.19 | 0.80 | 4.83 |
| ECGFiveDays | 0.54 | 0.92 | 0.95 | 0.89 | 1.32 | 0.90 | 5.45 |
| Lightning2 | 0.09 | 0.14 | 0.15 | 0.14 | 0.21 | 0.14 | 0.79 |
| MoteStrain | 0.76 | 1.29 | 1.34 | 1.23 | 1.90 | 1.30 | 7.88 |
| Plane | 0.13 | 0.22 | 0.23 | 0.21 | 0.31 | 0.21 | 1.25 |
| PowerCons | 0.23 | 0.39 | 0.40 | 0.36 | 0.55 | 0.37 | 2.20 |
| RefrigerationDevices | 0.48 | 0.85 | 0.88 | 0.80 | 1.19 | 0.83 | 5.01 |
| SonyAIBORobotSurface1 | 0.38 | 0.66 | 0.68 | 0.60 | 0.93 | 0.63 | 3.80 |
| SwedishLeaf | 0.71 | 1.21 | 1.24 | 1.13 | 1.73 | 1.16 | 6.95 |
| SyntheticControl | 0.37 | 0.63 | 0.65 | 0.62 | 0.92 | 0.61 | 3.74 |
| ToeSegmentation1 | 0.17 | 0.29 | 0.30 | 0.28 | 0.42 | 0.28 | 1.65 |
| Trace | 0.13 | 0.22 | 0.22 | 0.21 | 0.31 | 0.21 | 1.23 |
| TwoPatterns | 3.05 | 5.06 | 5.68 | 4.81 | 7.77 | 4.86 | 31.94 |
| Yoga | 1.91 | 3.47 | 3.58 | 3.26 | 4.85 | 3.35 | 21.55 |

## J COMPLETE EXPERIMENTAL RESULTS

Here, we provide complete experimental results with standard deviations on semi-supervised time-series classification presented in Section 4.1 and ablation studies in Section 4.2.

### J.1 SEMI-SUPERVISED TIME-SERIES CLASSIFICATION

We evaluated the model performance on semi-supervised time-series classification under both inductive and transductive inferences.

Table 7 depicts the classification performance of the proposed method compared to the baselines for various label ratios $\in [0.1, 0.9]$ under *inductive* inference. The proposed method achieved outstanding performance compared to the baselines on most datasets, especially with low label ratios. For example, in *SonyAIBORobotSurface1*, *SwedishLeaf*, *ToeSegmentation1*, and *Trace* datasets, MDTA improved classification performance by more than 4% than the second-best accuracy scores when

the label ratio was 0.1. These results demonstrate the effectiveness of MDTA in leveraging unlabeled instances in semi-supervised time-series classification under inductive inference.

In Table 8, we present the classification performance under *transductive* inference for MDTA and the baselines. The transductive inference evaluates model performance for unlabeled instances in the training dataset. Since MDTA focuses on enhancing the model's generalization performance on the test datasets, it may not perform well on transductive inference. Nevertheless, as shown in Table 8, MDTA and Pseudo perform better than the others under the transductive setting. Although the performance of the two methods is comparable, MDTA performs remarkably better than Pseudo in some datasets, such as *Lightning2*, *Trace*, and *TwoPatterns*, regardless of label ratios.

## J.2 ABLATION STUDIES

MDTA has three key components: *dual-temporal encoder*, *relation-preserving* loss function, and *random masking ratios*. To demonstrate their effectiveness, we compared MDTA to *MDTA w/o $f_1$*, *MDTA w/o $\mathcal{L}_{RP}$*, and *MDTA w/o RM*. Here, we used a fixed masking ratio of 0.5 for *MDTA w/o RM*.

Table 9 presents complete results on the ablation studies shown in Table 2 in Section 4.2. The performance of ablation models decreased compared to that of MDTA in most datasets. Especially for *MDTA w/o $f_1$*, which replaces the multi-resolution sub-encoder with one fully connected layer, showed remarkable performance degradation compared with the proposed method. Therefore, we demonstrate that each component of MDTA is essential to capture semantic information of time series effectively, enhancing model performance.

Table 7: Classification performance of baselines and MDTA for various label ratios $\in [0.1, 0.9]$ under *inductive* inference. The value in parentheses denotes the standard deviation. For each label ratio in datasets, the best score is highlighted in boldface. ($\mathbb{D}$: Dataset, LR: Label Ratio)

| $\mathbb{D}$ | LR | CE | Pseudo | Π-model | FixMatch | MTL | SSTSC | iTimes | MDTA |
|---|---|---|---|---|---|---|---|---|---|
| CB | 0.1 | 98.06 (1.11) | 98.92 (0.76) | 99.03 (0.99) | 98.60 (0.87) | 96.99 (0.73) | 98.82 (0.40) | 97.96 (1.61) | **99.78** (0.26) |
| | 0.2 | 98.82 (0.92) | 99.25 (0.55) | 98.71 (0.26) | **99.68** (0.43) | 97.63 (2.17) | 99.46 (0.59) | 99.03 (0.40) | 99.57 (0.40) |
| | 0.3 | 99.03 (0.71) | 98.92 (1.23) | 99.57 (0.40) | 99.03 (0.86) | 99.14 (0.94) | 99.46 (0.68) | **99.68** (0.26) | **99.68** (0.43) |
| | 0.4 | 99.25 (0.55) | 99.25 (0.80) | 99.46 (0.48) | **99.78** (0.43) | 98.92 (1.13) | 99.03 (1.94) | 99.68 (0.43) | **99.78** (0.26) |
| | 0.5 | 99.46 (0.68) | 99.03 (0.71) | 99.03 (0.71) | 99.46 (0.00) | 98.71 (1.47) | **99.68** (0.43) | 99.35 (0.63) | 99.57 (0.40) |
| | 0.6 | 99.46 (0.83) | 99.57 (0.40) | 99.46 (0.34) | 99.46 (0.83) | 99.03 (0.63) | 99.46 (0.34) | **99.78** (0.26) | 99.57 (0.22) |
| | 0.7 | **99.78** (0.26) | 99.14 (0.55) | 99.25 (0.43) | 99.46 (0.59) | 97.63 (1.75) | 99.57 (0.40) | 99.68 (0.43) | **99.78** (0.43) |
| | 0.8 | 99.25 (0.73) | 99.68 (0.65) | 99.46 (0.59) | 99.57 (0.40) | 99.68 (0.43) | 99.68 (0.26) | 99.78 (0.26) | **100.00** (0.00) |
| | 0.9 | 99.68 (0.43) | 99.89 (0.22) | 99.35 (0.53) | 99.57 (0.40) | 97.63 (1.82) | 99.25 (1.05) | **100.00** (0.00) | 99.68 (0.43) |
| CX | 0.1 | 34.10 (4.50) | 36.28 (4.86) | 47.44 (4.39) | 42.18 (3.30) | 27.56 (3.06) | 30.38 (5.14) | **49.87** (2.70) | 49.10 (4.82) |
| | 0.2 | 44.10 (5.07) | 42.44 (4.91) | 59.62 (1.62) | 52.05 (3.18) | 35.26 (2.40) | 37.56 (3.55) | **61.41** (3.96) | 48.97 (3.96) |
| | 0.3 | 44.49 (6.58) | 47.95 (2.20) | 57.31 (6.34) | 56.03 (4.34) | 34.87 (4.63) | 39.23 (3.33) | **63.72** (2.45) | 61.92 (3.68) |
| | 0.4 | 51.54 (4.93) | 51.54 (6.06) | 61.41 (3.43) | 59.49 (5.84) | 35.77 (5.52) | 42.05 (3.62) | **67.82** (3.15) | 65.64 (3.88) |
| | 0.5 | 57.69 (4.33) | 55.90 (1.96) | 63.97 (2.48) | 66.03 (3.69) | 40.90 (3.25) | 40.51 (3.66) | **69.74** (1.24) | 67.56 (2.35) |
| | 0.6 | 59.23 (1.50) | 58.21 (3.28) | 66.03 (2.40) | 66.15 (3.45) | 44.23 (6.41) | 43.08 (5.82) | **72.69** (4.54) | 68.97 (4.67) |
| | 0.7 | 57.05 (2.03) | 58.33 (3.27) | 67.18 (2.31) | 63.08 (2.83) | 42.05 (6.37) | 46.92 (6.48) | **72.95** (3.85) | 70.26 (4.01) |
| | 0.8 | 60.38 (1.37) | 62.69 (2.00) | 67.44 (4.18) | 64.74 (5.03) | 48.59 (1.79) | 52.31 (5.39) | **72.44** (4.75) | 71.41 (3.33) |
| | 0.9 | 61.79 (2.71) | 60.26 (2.92) | 66.92 (2.58) | 67.18 (1.96) | 54.23 (3.28) | 45.00 (8.27) | 72.69 (2.97) | **72.82** (1.32) |
| EF | 0.1 | 94.80 (0.97) | 96.72 (1.40) | 76.72 (5.89) | 75.37 (4.71) | 95.48 (1.38) | 95.71 (1.46) | 84.86 (3.70) | **99.55** (0.42) |
| | 0.2 | 98.64 (0.85) | 97.74 (1.19) | 83.28 (4.53) | 71.53 (2.25) | 99.19 (1.73) | 97.63 (2.30) | 93.79 (2.70) | **99.89** (0.23) |
| | 0.3 | 98.53 (1.11) | 98.53 (1.05) | 80.79 (4.73) | 79.55 (8.03) | 98.98 (0.42) | 97.85 (1.15) | 94.35 (3.52) | **99.89** (0.23) |
| | 0.4 | 99.32 (0.23) | 98.31 (1.01) | 78.53 (1.01) | 84.63 (3.06) | 98.98 (0.42) | 98.19 (0.97) | 93.45 (3.15) | **99.66** (0.45) |
| | 0.5 | 98.87 (0.71) | 98.53 (0.45) | 84.29 (6.26) | 82.71 (2.92) | 98.31 (1.24) | 98.98 (1.04) | 97.06 (0.97) | **99.77** (0.45) |
| | 0.6 | 98.64 (1.05) | 99.10 (0.28) | 79.10 (5.85) | 88.47 (3.89) | 99.32 (0.55) | 98.76 (0.90) | 98.19 (0.97) | **99.89** (0.23) |
| | 0.7 | 99.21 (0.58) | 99.21 (0.58) | 89.04 (2.22) | 86.21 (3.00) | 98.98 (0.42) | 99.32 (0.42) | 98.98 (0.90) | **99.66** (0.45) |
| | 0.8 | 99.32 (0.23) | 99.44 (0.36) | 89.38 (3.00) | 91.98 (3.18) | 98.53 (1.22) | 98.76 (0.66) | 98.76 (0.83) | **100.00** (0.00) |
| | 0.9 | 99.10 (0.28) | 98.98 (0.42) | 89.83 (3.57) | 89.49 (3.04) | 98.76 (0.66) | 98.98 (0.66) | 98.53 (0.92) | **100.00** (0.00) |
| L2 | 0.1 | 59.20 (6.88) | 60.80 (4.66) | 63.20 (3.92) | **64.80** (5.88) | 64.00 (6.69) | 63.20 (6.88) | **64.80** (7.76) | 61.60 (4.08) |
| | 0.2 | 62.40 (4.80) | 63.20 (4.66) | 62.40 (4.08) | 66.40 (7.42) | 62.40 (6.50) | 64.80 (2.99) | 72.80 (10.24) | **73.60** (4.08) |
| | 0.3 | 64.80 (2.99) | 70.40 (6.50) | 65.60 (7.42) | 68.00 (4.38) | 68.80 (4.66) | 68.00 (5.66) | **75.20** (6.88) | 66.40 (5.99) |
| | 0.4 | 69.60 (6.97) | 66.40 (4.80) | **72.00** (8.00) | 69.60 (8.24) | 66.40 (6.50) | 64.80 (6.40) | 71.20 (5.88) | 67.20 (13.00) |
| | 0.5 | 67.20 (5.88) | 68.80 (5.31) | 72.00 (6.69) | 73.60 (6.50) | 68.00 (4.38) | 68.80 (1.60) | 69.60 (5.43) | **76.80** (6.88) |
| | 0.6 | 67.20 (6.40) | 72.00 (4.38) | 67.20 (6.40) | 71.20 (9.26) | 65.60 (6.97) | 68.80 (2.99) | 67.20 (5.31) | **85.60** (3.20) |
| | 0.7 | 72.80 (4.66) | 69.60 (10.31) | 72.00 (8.39) | 72.00 (9.80) | 70.40 (5.43) | 66.40 (5.99) | 76.00 (6.69) | **84.30** (2.53) |
| | 0.8 | 73.60 (6.97) | 76.80 (4.66) | 76.80 (9.93) | 76.00 (5.06) | 73.60 (5.43) | 72.80 (3.92) | 73.60 (6.97) | **78.40** (5.99) |
| | 0.9 | 71.20 (1.60) | 72.00 (2.53) | 69.60 (5.43) | 67.20 (4.66) | 68.80 (2.99) | 71.20 (4.66) | 73.60 (8.24) | **83.20** (6.88) |
| MS | 0.1 | 88.78 (1.82) | 89.73 (3.08) | **91.84** (2.51) | 90.35 (1.54) | 86.20 (3.36) | 88.31 (1.76) | 89.88 (1.63) | 91.14 (0.95) |
| | 0.2 | 92.55 (1.26) | 90.59 (2.57) | 93.49 (1.13) | 93.96 (1.69) | 87.61 (2.13) | 91.61 (2.07) | 93.41 (1.17) | **95.29** (0.43) |
| | 0.3 | 91.76 (1.53) | 93.33 (1.92) | 93.49 (0.95) | 94.35 (0.88) | 87.92 (2.53) | 91.37 (2.00) | 93.33 (1.24) | **95.22** (1.00) |
| | 0.4 | 93.49 (1.60) | 93.18 (1.37) | 93.88 (0.84) | 94.82 (0.76) | 89.65 (2.90) | 91.45 (1.80) | 93.88 (0.81) | **95.14** (1.60) |
| | 0.5 | 93.18 (1.37) | 94.04 (1.20) | 94.98 (1.83) | **95.22** (1.09) | 93.25 (0.80) | 93.65 (1.17) | 94.20 (1.73) | 95.06 (1.01) |
| | 0.6 | 93.80 (1.30) | 95.29 (0.66) | 94.27 (1.50) | 94.27 (1.66) | 90.67 (1.74) | 93.41 (3.08) | 93.73 (0.89) | **95.37** (1.56) |
| | 0.7 | 94.98 (0.67) | 94.98 (0.84) | 95.29 (0.82) | 94.20 (1.50) | 84.16 (9.14) | 94.04 (0.84) | 93.96 (1.10) | **96.24** (0.64) |
| | 0.8 | 95.45 (0.91) | 94.82 (0.63) | 95.69 (0.96) | 95.06 (0.68) | 92.47 (1.83) | 92.16 (1.31) | 94.75 (0.40) | **95.69** (0.61) |
| | 0.9 | 95.61 (0.94) | 95.22 (1.06) | 94.43 (2.12) | 95.45 (0.53) | 90.59 (4.21) | 94.75 (1.62) | 94.90 (0.86) | **96.08** (0.61) |

| | | | | | | | | | |
|---|---|---|---|---|---|---|---|---|---|
| PL | 0.1 | 93.33 (4.10) | 91.43 (6.49) | 53.33 (14.17) | 62.86 (10.17) | 63.81 (11.41) | 85.71 (6.90) | 63.33 (12.47) | **95.71** (0.95) |
| | 0.2 | **96.19** (2.43) | 94.76 (2.33) | 80.00 (9.83) | 75.71 (8.30) | 77.14 (6.14) | 92.86 (3.98) | 69.52 (12.90) | 95.71 (1.78) |
| | 0.3 | 95.71 (2.33) | **96.67** (1.90) | 82.86 (9.21) | 88.10 (7.97) | 85.24 (11.31) | 92.38 (3.50) | 81.90 (10.17) | 95.71 (1.78) |
| | 0.4 | 97.62 (1.51) | **98.10** (1.78) | 88.10 (9.04) | 92.38 (7.59) | 91.43 (5.13) | 94.29 (3.87) | 89.05 (8.33) | 95.71 (1.78) |
| | 0.5 | 96.67 (1.17) | **98.10** (1.78) | 87.62 (6.97) | 96.19 (3.23) | 86.19 (10.15) | 96.67 (2.43) | 93.81 (2.43) | 96.67 (1.90) |
| | 0.6 | 97.62 (1.51) | **98.57** (1.17) | 93.81 (8.98) | 95.71 (4.36) | 90.95 (8.16) | 93.81 (4.67) | 95.24 (1.51) | 97.62 (1.51) |
| | 0.7 | **98.57** (1.17) | 97.62 (1.51) | 93.33 (3.81) | 93.33 (5.91) | 90.95 (5.91) | 97.62 (1.51) | 84.29 (8.05) | 96.67 (2.43) |
| | 0.8 | 98.57 (1.17) | 98.10 (0.95) | 95.24 (4.76) | 96.19 (4.15) | 86.67 (9.11) | 98.10 (2.33) | 97.14 (0.95) | **99.05** (1.17) |
| | 0.9 | 98.57 (1.17) | **99.52** (0.95) | **99.52** (0.95) | 98.10 (1.78) | 91.43 (5.35) | 99.05 (1.17) | 96.67 (3.23) | 99.05 (1.90) |
| PC | 0.1 | 85.00 (3.22) | 86.94 (3.36) | 84.44 (1.62) | **87.50** (1.96) | 84.17 (4.08) | 86.39 (1.62) | 85.56 (3.12) | 86.67 (2.26) |
| | 0.2 | 85.00 (4.16) | 85.83 (2.22) | 86.11 (1.76) | 85.83 (2.04) | 86.67 (1.67) | 88.33 (1.42) | 86.39 (3.22) | **91.67** (1.76) |
| | 0.3 | 88.33 (2.08) | 87.22 (2.39) | 84.17 (0.68) | 84.72 (1.96) | 88.06 (3.36) | 88.61 (1.62) | 87.22 (2.04) | **93.33** (1.62) |
| | 0.4 | 88.89 (3.73) | 87.22 (2.83) | 84.72 (1.52) | 85.83 (2.39) | 88.89 (1.96) | 88.33 (1.88) | 87.78 (1.84) | **90.56** (1.62) |
| | 0.5 | 90.56 (1.36) | 89.72 (1.88) | 86.67 (2.26) | 87.22 (1.04) | 86.94 (3.24) | 90.83 (2.08) | 87.78 (2.04) | **94.44** (2.91) |
| | 0.6 | 91.11 (1.88) | 91.67 (1.96) | 86.67 (2.08) | 85.83 (1.62) | 87.78 (2.83) | 89.72 (3.58) | 87.22 (2.39) | **94.44** (1.24) |
| | 0.7 | 90.83 (0.68) | 89.72 (1.88) | 85.28 (1.88) | 85.83 (1.36) | 87.78 (4.60) | 88.06 (2.08) | 88.33 (1.42) | **95.56** (1.84) |
| | 0.8 | 92.78 (2.39) | 89.72 (1.67) | 83.89 (1.42) | 88.06 (1.67) | 91.11 (1.42) | 90.28 (1.96) | 90.28 (1.24) | **97.22** (0.88) |
| | 0.9 | 92.22 (1.88) | 91.94 (2.22) | 86.39 (1.04) | 85.28 (1.11) | 89.17 (5.08) | 91.11 (2.26) | 89.44 (2.08) | **98.33** (0.56) |
| RD | 0.1 | 54.00 (4.42) | 52.40 (4.12) | 52.00 (3.38) | 51.07 (2.25) | 56.00 (4.46) | **56.13** (4.53) | 52.27 (4.31) | 53.60 (2.09) |
| | 0.2 | 56.67 (3.07) | 55.73 (1.40) | 56.00 (3.64) | 55.33 (1.74) | 56.00 (2.49) | 57.20 (2.08) | 57.60 (3.71) | **58.00** (3.65) |
| | 0.3 | 57.60 (1.55) | 56.27 (2.84) | 56.13 (2.59) | 57.87 (2.96) | 54.27 (2.00) | 56.40 (2.82) | **58.13** (2.36) | 57.07 (2.29) |
| | 0.4 | 55.47 (4.18) | 56.13 (3.04) | 58.53 (1.52) | 55.87 (2.25) | 58.00 (1.12) | 59.20 (3.56) | **59.87** (2.61) | 58.27 (3.20) |
| | 0.5 | 59.73 (1.87) | 59.33 (4.75) | 57.20 (3.80) | 60.67 (3.77) | 57.73 (3.90) | 58.00 (4.68) | **61.87** (2.36) | 61.07 (1.16) |
| | 0.6 | 60.61 (2.80) | 57.20 (4.74) | 55.60 (1.19) | 58.00 (2.27) | 58.00 (1.84) | 55.33 (3.18) | **62.80** (2.04) | 60.40 (3.64) |
| | 0.7 | 60.80 (2.99) | 59.47 (3.52) | 58.67 (3.86) | 61.73 (1.61) | 58.93 (4.35) | 59.07 (4.31) | **62.40** (0.90) | 61.73 (3.26) |
| | 0.8 | 61.20 (1.29) | 58.93 (3.60) | 58.13 (3.19) | 58.00 (0.73) | 59.47 (1.95) | 60.53 (3.71) | **64.67** (6.15) | 62.27 (4.10) |
| | 0.9 | 61.33 (2.49) | 63.07 (1.67) | 61.07 (1.74) | 57.33 (0.73) | 59.20 (3.30) | 61.87 (1.54) | **66.13** (1.76) | 63.60 (2.82) |
| SR | 0.1 | 95.36 (2.50) | 94.08 (0.82) | 92.64 (1.55) | 92.96 (0.60) | 92.96 (2.65) | 95.36 (1.38) | 92.16 (0.78) | **99.52** (0.39) |
| | 0.2 | 96.80 (1.34) | 96.80 (2.37) | 92.48 (1.48) | 94.72 (0.82) | 93.92 (1.20) | 95.84 (0.93) | 92.32 (2.18) | **99.36** (0.60) |
| | 0.3 | 98.24 (1.18) | 96.96 (1.28) | 94.72 (1.04) | 94.56 (1.85) | 92.96 (3.52) | 96.48 (0.82) | 93.60 (1.82) | **99.04** (0.93) |
| | 0.4 | 96.96 (1.55) | 96.32 (0.82) | 93.12 (1.09) | 93.92 (1.09) | 95.36 (1.28) | 97.12 (0.82) | 93.60 (2.26) | **99.52** (0.64) |
| | 0.5 | 98.08 (0.39) | 98.08 (1.20) | 93.60 (1.13) | 95.36 (1.55) | 96.64 (0.32) | 96.48 (0.64) | 95.04 (1.38) | **99.84** (0.32) |
| | 0.6 | 97.76 (1.06) | 98.24 (1.28) | 93.44 (1.55) | 94.40 (1.01) | 93.76 (3.87) | 97.60 (0.88) | 95.68 (1.57) | **99.52** (0.64) |
| | 0.7 | 97.60 (0.72) | 98.40 (0.88) | 93.12 (1.30) | 94.56 (1.63) | 87.52 (15.78) | 97.76 (1.06) | 95.68 (0.82) | **100.00** (0.00) |
| | 0.8 | 97.28 (1.20) | 97.92 (1.20) | 94.24 (1.47) | 93.12 (1.09) | 90.56 (13.00) | 97.44 (1.18) | 94.56 (1.78) | **99.84** (0.32) |
| | 0.9 | 98.08 (0.64) | 98.08 (0.96) | 93.60 (2.21) | 94.40 (1.01) | 95.84 (1.18) | 97.12 (1.09) | 95.20 (1.89) | **99.84** (0.32) |
| SL | 0.1 | 63.64 (5.72) | 70.22 (3.13) | 52.36 (9.53) | 51.56 (7.16) | 41.87 (3.50) | 57.78 (6.94) | 44.44 (3.96) | **74.67** (3.80) |
| | 0.2 | 75.02 (2.47) | 76.27 (1.34) | 69.24 (2.67) | 65.69 (4.17) | 43.29 (4.40) | 66.67 (5.09) | 46.13 (3.53) | **76.62** (3.43) |
| | 0.3 | 83.64 (1.67) | 82.22 (0.84) | 67.11 (3.18) | 66.31 (4.52) | 55.82 (10.28) | 75.47 (3.64) | 53.16 (5.31) | **85.07** (3.09) |
| | 0.4 | 85.96 (2.81) | 84.53 (4.34) | 70.40 (4.47) | 71.11 (3.01) | 66.13 (10.21) | 81.60 (2.87) | 57.16 (2.51) | **87.73** (1.55) |
| | 0.5 | 87.56 (2.28) | **88.00** (1.97) | 71.82 (1.68) | 71.91 (4.81) | 54.76 (10.12) | 84.18 (4.25) | 56.71 (4.20) | 87.29 (1.24) |
| | 0.6 | 89.87 (1.39) | 88.62 (1.72) | 74.49 (3.42) | 71.02 (3.87) | 55.82 (13.74) | 80.53 (2.15) | 60.98 (4.14) | **89.96** (1.79) |
| | 0.7 | 90.31 (1.55) | 91.20 (1.30) | 73.07 (1.70) | 74.22 (5.64) | 58.84 (10.11) | 81.69 (7.88) | 62.49 (5.48) | **92.00** (1.57) |
| | 0.8 | 89.51 (2.49) | **90.84** (2.83) | 75.73 (6.00) | 77.87 (3.91) | 58.93 (6.15) | 83.11 (3.45) | 62.76 (5.22) | 90.84 (2.22) |
| | 0.9 | **91.91** (2.29) | 91.56 (1.23) | 78.84 (2.00) | 77.24 (4.89) | 63.56 (6.82) | 81.33 (4.39) | 61.78 (5.75) | 91.47 (1.79) |
| SC | 0.1 | 88.00 (5.23) | 95.67 (2.20) | 95.67 (1.78) | 95.50 (1.35) | 92.17 (1.25) | 86.50 (5.44) | 92.67 (3.99) | **96.67** (2.11) |
| | 0.2 | 92.33 (4.20) | 95.83 (4.01) | **97.83** (0.85) | 97.67 (0.62) | 96.83 (0.82) | 94.33 (3.47) | 95.17 (1.43) | 97.00 (1.35) |
| | 0.3 | 96.83 (1.93) | 95.50 (1.63) | 97.33 (0.82) | 96.67 (1.83) | 96.33 (2.51) | 93.00 (2.72) | 96.83 (2.32) | **99.00** (0.62) |
| | 0.4 | 97.67 (1.70) | 98.33 (1.39) | **98.67** (1.94) | 98.50 (1.43) | 97.33 (2.07) | 93.67 (2.33) | 98.17 (1.33) | 98.67 (0.85) |
| | 0.5 | 97.17 (1.25) | **98.50** (0.97) | 97.33 (1.11) | 97.50 (1.39) | 98.00 (1.25) | 94.50 (2.51) | 98.33 (0.53) | 98.00 (0.85) |
| | 0.6 | **99.33** (0.62) | 98.17 (1.11) | 97.67 (1.33) | 97.83 (1.87) | 97.50 (1.05) | 95.67 (2.26) | 98.00 (1.13) | 97.67 (1.11) |
| | 0.7 | 98.17 (1.78) | **99.50** (0.41) | 99.17 (1.29) | 96.50 (1.78) | 98.17 (1.53) | 95.33 (1.80) | 97.83 (1.13) | 98.50 (0.97) |
| | 0.8 | 98.17 (0.62) | 99.00 (0.62) | **99.33** (0.62) | 98.50 (0.62) | 98.33 (0.91) | 95.50 (2.61) | 99.17 (0.00) | 98.17 (1.22) |
| | 0.9 | 98.00 (1.13) | 98.83 (1.13) | 98.83 (1.13) | 99.00 (1.22) | 98.17 (1.43) | 95.50 (2.21) | **99.67** (0.41) | 99.33 (0.62) |
| T1 | 0.1 | 68.15 (7.63) | 71.85 (6.35) | 68.89 (9.76) | 72.59 (6.24) | 72.22 (7.12) | 69.63 (5.57) | 69.63 (4.63) | **82.59** (1.89) |
| | 0.2 | 75.19 (4.32) | 77.41 (7.35) | 77.41 (5.54) | 80.74 (7.91) | 77.41 (8.06) | 73.33 (8.41) | 83.33 (3.31) | **89.26** (4.12) |
| | 0.3 | 81.48 (4.68) | 82.59 (4.77) | 81.85 (7.63) | 80.37 (2.77) | 73.33 (6.69) | 81.85 (5.79) | 81.85 (4.12) | **96.67** (2.16) |
| | 0.4 | 87.41 (2.96) | 84.44 (4.77) | 83.33 (5.62) | 82.22 (8.25) | 84.44 (3.81) | 85.19 (3.31) | 87.41 (2.72) | **95.56** (2.22) |
| | 0.5 | 86.30 (6.37) | 84.81 (6.56) | 89.26 (2.46) | 89.63 (3.23) | 87.78 (4.77) | 85.56 (6.97) | 88.89 (4.83) | 94.81 (2.96) |
| | 0.6 | 88.89 (4.54) | 85.56 (4.29) | 90.37 (3.19) | 85.56 (4.29) | 81.48 (4.22) | 82.96 (4.60) | 88.89 (5.11) | **97.04** (1.48) |
| | 0.7 | 90.00 (3.01) | 90.37 (1.39) | 91.11 (4.29) | 89.26 (3.59) | 86.67 (4.29) | 86.30 (5.19) | 91.85 (2.77) | **97.04** (2.77) |
| | 0.8 | 88.89 (2.62) | 91.85 (3.01) | 90.37 (1.81) | 89.26 (3.59) | 87.04 (4.06) | 88.52 (5.16) | 94.44 (3.10) | **96.30** (3.10) |
| | 0.9 | 91.11 (2.46) | 91.11 (2.16) | 90.00 (3.23) | 91.48 (2.51) | 90.37 (2.96) | 88.15 (3.63) | 94.07 (1.39) | **97.04** (1.48) |
| TR | 0.1 | 75.50 (10.42) | 72.00 (11.34) | 74.50 (3.67) | 80.50 (6.78) | 75.50 (5.10) | 67.50 (8.22) | 81.50 (8.46) | **90.00** (10.25) |
| | 0.2 | 84.00 (6.24) | 92.50 (5.70) | 83.50 (5.83) | 89.00 (8.28) | 81.50 (8.46) | 86.00 (9.70) | 96.00 (3.39) | **99.50** (1.00) |
| | 0.3 | 93.50 (4.06) | 94.50 (6.20) | 95.50 (6.00) | 92.00 (9.92) | 90.50 (4.30) | 94.50 (4.30) | 95.50 (3.32) | **99.00** (1.22) |
| | 0.4 | 94.00 (2.55) | 94.50 (6.60) | 95.00 (7.58) | 91.50 (6.63) | 94.50 (4.00) | 87.00 (11.00) | 97.00 (2.92) | **99.50** (1.00) |
| | 0.5 | 95.00 (3.87) | 97.50 (2.74) | 93.00 (5.34) | 94.00 (5.61) | 92.50 (7.25) | 96.00 (4.06) | 99.50 (1.00) | **100.00** (0.00) |
| | 0.6 | 96.50 (2.55) | 95.00 (2.74) | 96.50 (2.55) | 96.50 (1.22) | 94.50 (9.80) | 97.50 (2.74) | 97.50 (2.24) | **100.00** (0.00) |
| | 0.7 | 93.00 (5.10) | 98.00 (1.87) | 97.50 (1.58) | 96.50 (2.00) | 99.50 (1.00) | 97.00 (2.45) | 96.50 (2.00) | 99.00 (2.00) |
| | 0.8 | 97.00 (2.45) | 98.50 (2.00) | 93.50 (5.15) | 96.50 (2.00) | 98.50 (2.00) | 98.50 (2.24) | 99.00 (1.22) | **100.00** (0.00) |
| | 0.9 | 99.00 (1.22) | 96.50 (3.74) | 96.50 (2.55) | 97.50 (3.87) | 96.50 (5.83) | 99.00 (1.22) | **99.50** (1.00) | 99.00 (1.22) |
| TP | 0.1 | 99.30 (0.40) | 99.60 (0.32) | 98.98 (0.50) | 98.94 (0.42) | 96.88 (1.66) | 99.06 (0.72) | 93.28 (3.15) | **99.94** (0.08) |
| | 0.2 | 99.76 (0.29) | 99.86 (0.15) | 99.00 (0.30) | 99.46 (0.14) | 98.56 (0.57) | 99.72 (0.22) | 95.34 (1.08) | **99.94** (0.08) |
| | 0.3 | 99.90 (0.11) | 99.74 (0.16) | 99.32 (0.41) | 99.42 (0.15) | 98.72 (1.55) | 99.82 (0.12) | 96.92 (0.80) | **100.00** (0.00) |
| | 0.4 | 99.86 (0.15) | 99.84 (0.27) | 99.44 (0.33) | 99.20 (0.18) | 98.78 (0.75) | 99.82 (0.12) | 96.88 (0.44) | **100.00** (0.00) |
| | 0.5 | 99.88 (0.07) | 99.90 (0.15) | 99.22 (0.44) | 99.50 (0.46) | 99.66 (0.31) | 99.82 (0.07) | 97.62 (1.06) | **99.96** (0.08) |
| | 0.6 | 99.88 (0.07) | 99.90 (0.13) | 99.42 (0.18) | 99.70 (0.32) | 98.84 (1.31) | 99.78 (0.24) | 98.20 (0.32) | **99.96** (0.05) |
| | 0.7 | 99.92 (0.07) | 99.96 (0.05) | 99.36 (0.21) | 99.60 (0.30) | 99.10 (0.96) | 99.84 (0.05) | 97.74 (0.92) | **100.00** (0.00) |
| | 0.8 | 99.82 (0.07) | 99.94 (0.08) | 99.36 (0.57) | 99.76 (0.20) | 99.86 (0.08) | 99.90 (0.20) | 97.78 (0.54) | **99.96** (0.05) |
| | 0.9 | 99.94 (0.08) | 99.90 (0.06) | 99.62 (0.24) | 99.76 (0.08) | 99.78 (0.15) | 99.84 (0.21) | 98.02 (0.59) | **100.00** (0.00) |
| YO | 0.1 | **75.36** (1.07) | 72.76 (1.80) | 62.12 (1.34) | 61.58 (3.08) | 66.21 (2.47) | 73.70 (1.91) | 68.97 (1.83) | 73.48 (5.14) |
| | 0.2 | 77.36 (2.69) | 78.36 (1.85) | 63.36 (0.57) | 63.30 (3.10) | 69.79 (1.24) | 77.30 (0.94) | 71.61 (2.86) | **78.45** (5.09) |
| | 0.3 | **82.18** (2.00) | 81.64 (0.52) | 65.36 (2.16) | 61.94 (2.50) | 72.94 (4.46) | 80.36 (1.63) | 74.15 (1.95) | 81.73 (6.28) |
| | 0.4 | 83.55 (1.28) | 84.21 (1.61) | 65.03 (3.93) | 61.64 (0.94) | 76.21 (3.59) | 79.18 (2.94) | 71.30 (1.78) | **86.03** (5.85) |
| | 0.5 | 86.12 (1.02) | 85.64 (1.18) | 65.39 (2.08) | 64.33 (1.47) | 79.00 (1.27) | 81.76 (3.62) | 77.94 (2.18) | **89.36** (0.65) |

| | | | | | | | | |
|---|---|---|---|---|---|---|---|---|
| 0.6 | 86.85 (1.61) | 87.24 (1.12) | 66.88 (6.83) | 62.21 (2.33) | 77.15 (4.57) | 81.15 (4.57) | 79.39 (1.99) | **88.03** (4.09) |
| 0.7 | 86.94 (0.86) | 88.24 (1.26) | 64.18 (2.68) | 65.82 (3.01) | 76.58 (2.95) | 84.24 (2.11) | 78.88 (0.87) | **88.33** (7.22) |
| 0.8 | 88.52 (1.57) | 89.06 (0.65) | 65.06 (0.82) | 65.70 (3.22) | 75.73 (8.45) | 83.52 (1.70) | 80.03 (2.22) | **92.27** (2.88) |
| 0.9 | **89.03** (0.82) | 88.64 (1.21) | 65.12 (1.76) | 68.12 (5.30) | 79.27 (1.98) | 84.00 (2.23) | 81.00 (3.95) | 88.82 (4.23) |

Table 8: Classification performance of baselines and MDTA for various label ratios $\in [0.1, 0.9]$ under *transductive* inference. The value in parentheses denotes the standard deviation. For each label ratio in datasets, the best score is highlighted in boldface. ($\mathbb{D}$: Dataset, LR: Label Ratio)

| $\mathbb{D}$ | LR | CE | Pseudo | Π-model | FixMatch | MTL | SSTSC | iTimes | MDTA |
|---|---|---|---|---|---|---|---|---|---|
| CB | 0.1 | 98.01 (0.93) | 98.76 (0.77) | 98.88 (0.86) | 98.37 (0.90) | 96.06 (1.35) | 98.73 (0.73) | 97.97 (0.96) | **99.48** (0.37) |
| | 0.2 | 98.83 (0.17) | 98.88 (0.39) | 98.88 (0.38) | 99.33 (0.25) | 98.83 (1.46) | **99.42** (0.18) | 99.19 (0.46) | 99.27 (0.63) |
| | 0.3 | 99.44 (0.38) | 99.79 (0.38) | 99.64 (0.26) | 99.49 (0.56) | 99.13 (0.64) | 99.79 (0.19) | 99.74 (0.16) | **99.90** (0.41) |
| | 0.4 | 99.16 (0.74) | 99.58 (0.38) | 98.98 (0.77) | 99.58 (0.24) | 98.86 (1.22) | 98.98 (1.15) | **99.76** (0.29) | 99.64 (0.48) |
| | 0.5 | 99.14 (0.87) | **99.86** (0.18) | 99.21 (0.53) | 99.64 (0.39) | 98.85 (1.42) | **99.86** (0.18) | 99.14 (0.49) | 99.48 (0.38) |
| | 0.6 | 99.55 (0.49) | 99.82 (0.34) | 99.82 (0.22) | 99.64 (0.44) | 99.73 (0.36) | **99.91** (0.18) | **99.91** (0.15) | **99.91** (0.18) |
| | 0.7 | **99.88** (0.24) | 99.52 (0.29) | 99.64 (0.29) | **99.88** (0.24) | 99.28 (0.88) | 99.64 (0.29) | 99.40 (0.38) | 99.50 (0.73) |
| | 0.8 | **100.00** (0.00) | 99.82 (0.44) | 99.64 (0.44) | 99.28 (1.05) | 99.82 (0.36) | **100.00** (0.00) | **100.00** (0.00) | 99.64 (0.45) |
| | 0.9 | 99.64 (0.73) | **100.00** (0.00) | 98.18 (1.15) | 98.18 (1.15) | 97.09 (3.37) | **100.00** (0.00) | 99.64 (0.73) | **100.00** (0.00) |
| CX | 0.1 | 41.28 (3.65) | 40.14 (4.46) | 49.45 (4.68) | 43.71 (1.43) | 32.26 (2.49) | 36.86 (4.17) | **53.87** (3.10) | 49.62 (4.10) |
| | 0.2 | 54.39 (5.92) | 53.53 (4.63) | 60.86 (2.15) | 55.35 (2.31) | 44.39 (1.59) | 45.13 (4.71) | **64.92** (4.03) | 53.14 (0.81) |
| | 0.3 | 57.31 (5.81) | 60.68 (3.89) | 61.47 (5.34) | 60.55 (6.14) | 50.95 (5.69) | 54.07 (4.67) | **71.87** (3.24) | 64.50 (0.00) |
| | 0.4 | 67.71 (5.59) | 67.29 (2.06) | 65.14 (3.71) | 63.86 (6.05) | 50.07 (6.83) | 58.57 (3.24) | **76.57** (3.74) | 69.43 (2.22) |
| | 0.5 | 73.08 (9.92) | 70.43 (1.75) | 69.15 (1.41) | 68.80 (0.72) | 59.06 (3.87) | 56.32 (7.61) | **76.58** (2.15) | 74.78 (2.35) |
| | 0.6 | 78.18 (9.97) | 78.40 (3.84) | 71.34 (3.78) | 71.02 (3.80) | 63.21 (13.95) | 60.53 (10.32) | **80.64** (2.98) | 74.89 (2.34) |
| | 0.7 | 80.14 (13.82) | 80.71 (3.54) | 69.43 (3.45) | 67.86 (5.59) | 58.43 (13.86) | 69.14 (9.05) | **84.71** (2.29) | 74.57 (2.29) |
| | 0.8 | 86.02 (11.95) | **87.10** (2.26) | 72.90 (4.78) | 71.61 (6.51) | 67.10 (2.77) | 73.12 (7.13) | 85.81 (5.33) | 78.44 (1.33) |
| | 0.9 | 88.70 (11.03) | **89.57** (2.95) | 70.43 (1.74) | 70.87 (3.53) | 78.26 (9.33) | 72.61 (16.24) | 84.78 (3.07) | 80.50 (3.67) |
| EF | 0.1 | 95.56 (1.10) | 97.32 (0.78) | 82.81 (2.39) | 82.52 (2.12) | 97.53 (1.52) | 96.77 (1.58) | 87.92 (2.19) | **99.40** (0.25) |
| | 0.2 | 98.96 (0.46) | 99.01 (0.28) | 87.31 (3.69) | 79.34 (2.21) | 98.87 (0.50) | 98.11 (1.79) | 96.46 (1.75) | **99.86** (0.19) |
| | 0.3 | 99.51 (0.52) | 99.14 (0.36) | 85.82 (3.93) | 85.28 (5.64) | 99.46 (0.24) | 98.44 (1.39) | 95.63 (2.53) | **99.68** (0.38) |
| | 0.4 | **99.94** (0.13) | 99.43 (1.19) | 82.70 (0.60) | 87.23 (4.31) | 99.25 (0.62) | 98.55 (1.04) | 95.72 (2.91) | 99.68 (0.35) |
| | 0.5 | **99.92** (0.15) | 99.70 (0.53) | 88.23 (3.75) | 85.89 (2.21) | 99.47 (0.51) | 99.55 (0.73) | 97.36 (1.86) | 99.77 (0.19) |
| | 0.6 | 99.34 (0.64) | 99.62 (0.35) | 83.68 (3.41) | 90.66 (2.91) | **99.72** (0.38) | 99.43 (0.69) | 99.34 (0.57) | 99.62 (0.36) |
| | 0.7 | 99.75 (0.50) | 99.87 (0.31) | 93.71 (1.87) | 91.57 (1.81) | 99.75 (0.50) | 99.37 (0.40) | 99.50 (0.47) | **100.00** (0.00) |
| | 0.8 | 99.81 (0.38) | **100.00** (0.00) | 92.83 (2.77) | 93.02 (2.96) | 99.25 (0.38) | 99.06 (1.19) | 98.11 (1.58) | **100.00** (0.00) |
| | 0.9 | **100.00** (0.00) | **100.00** (0.00) | 90.94 (5.65) | 94.34 (2.39) | **100.00** (0.00) | 99.25 (1.11) | 98.11 (1.19) | **100.00** (0.00) |
| L2 | 0.1 | 61.56 (8.82) | 68.13 (5.85) | 65.94 (4.98) | **68.75** (7.57) | 63.44 (8.98) | 68.44 (4.24) | 66.56 (6.96) | 64.33 (4.67) |
| | 0.2 | 67.02 (3.22) | 69.47 (6.96) | 69.74 (2.28) | 65.61 (5.72) | 69.12 (2.85) | 66.32 (5.59) | 67.54 (6.14) | **72.50** (5.36) |
| | 0.3 | 72.40 (4.08) | 71.50 (4.56) | 68.80 (6.14) | 67.20 (0.98) | 74.40 (4.08) | 70.80 (3.25) | 72.80 (10.93) | **76.50** (5.72) |
| | 0.4 | **78.14** (5.42) | 75.81 (6.41) | 65.12 (9.53) | 68.84 (8.00) | 74.88 (3.42) | 73.02 (2.37) | 72.09 (2.55) | 70.83 (17.00) |
| | 0.5 | 79.44 (4.16) | 77.22 (4.61) | 64.44 (5.39) | 68.89 (7.74) | **83.33** (3.93) | 72.78 (4.08) | 77.78 (4.97) | 78.00 (5.42) |
| | 0.6 | 78.57 (6.68) | 77.14 (12.90) | 62.14 (7.35) | 67.86 (3.19) | 73.21 (3.99) | 75.71 (6.93) | 79.29 (4.16) | **89.00** (5.83) |
| | 0.7 | 86.67 (15.47) | 75.24 (11.51) | 69.52 (9.81) | 65.71 (8.19) | 85.71 (5.22) | 81.90 (9.23) | 77.14 (4.67) | **88.00** (6.00) |
| | 0.8 | 87.14 (15.25) | 67.14 (7.28) | 72.86 (13.09) | 75.71 (7.28) | **91.43** (8.33) | 78.57 (12.78) | 85.71 (10.10) | 88.00 (4.00) |
| | 0.9 | 80.00 (11.43) | 80.00 (10.69) | 60.00 (10.69) | 68.57 (16.66) | 80.00 (7.00) | **92.86** (7.14) | 80.00 (11.43) | 84.00 (14.97) |
| MS | 0.1 | 91.15 (1.62) | 91.74 (1.99) | **92.61** (1.20) | 91.74 (1.06) | 89.81 (1.43) | 90.36 (1.24) | 91.18 (1.11) | 90.74 (0.91) |
| | 0.2 | 93.63 (1.16) | 94.22 (1.40) | 93.46 (0.73) | 94.22 (1.69) | 90.74 (1.94) | 93.56 (1.19) | 93.96 (1.46) | **94.63** (1.46) |
| | 0.3 | 95.20 (1.18) | 94.37 (0.87) | 94.78 (0.71) | 94.78 (0.59) | 93.10 (0.48) | 94.07 (1.90) | 94.15 (0.74) | **95.77** (0.09) |
| | 0.4 | 95.58 (0.91) | 95.01 (0.91) | 94.70 (0.58) | 95.01 (1.52) | 92.43 (2.23) | 95.54 (0.92) | 95.10 (1.45) | **96.09** (0.68) |
| | 0.5 | 96.64 (1.22) | 96.59 (0.45) | 94.96 (1.65) | 95.64 (0.73) | 95.85 (0.67) | 96.59 (0.80) | 95.91 (0.95) | **97.68** (0.61) |
| | 0.6 | **96.97** (1.46) | 96.71 (0.83) | 95.79 (0.98) | 94.93 (2.32) | 92.96 (2.84) | 95.39 (2.71) | 94.74 (1.43) | 95.93 (0.25) |
| | 0.7 | 96.67 (1.85) | **97.28** (1.66) | 96.05 (0.73) | 94.04 (2.84) | 92.00 (5.67) | 96.93 (1.14) | 95.35 (0.59) | 95.82 (0.73) |
| | 0.8 | 98.03 (2.00) | **98.16** (0.76) | 95.00 (2.23) | 94.21 (1.83) | 95.79 (1.35) | 95.92 (1.92) | 96.32 (1.59) | 95.47 (0.98) |
| | 0.9 | **98.68** (2.04) | 97.89 (1.34) | 95.79 (2.11) | 96.32 (0.98) | 95.53 (3.49) | 98.42 (1.53) | 96.32 (2.11) | 96.29 (1.14) |
| PL | 0.1 | 93.81 (2.31) | 90.09 (6.56) | 55.58 (11.06) | 62.48 (5.89) | 66.73 (8.39) | 89.56 (3.03) | 63.19 (12.35) | **96.00** (0.93) |
| | 0.2 | 95.60 (2.94) | 96.40 (2.00) | 78.60 (9.26) | 75.20 (8.75) | 80.20 (7.22) | 93.20 (2.56) | 71.80 (11.16) | **97.40** (3.26) |
| | 0.3 | 97.73 (1.02) | **99.09** (1.33) | 82.05 (8.49) | 87.05 (9.90) | 88.41 (9.71) | 94.77 (2.34) | 84.55 (12.67) | 97.50 (1.94) |
| | 0.4 | 97.87 (1.07) | **98.93** (0.53) | 89.87 (6.98) | 91.73 (6.61) | 92.27 (5.16) | 97.60 (1.55) | 92.53 (3.64) | 96.29 (1.94) |
| | 0.5 | 98.10 (1.19) | **99.37** (0.78) | 86.03 (7.34) | 95.24 (3.33) | 92.06 (11.96) | 96.83 (1.74) | 91.43 (3.56) | 96.00 (2.00) |
| | 0.6 | **99.60** (0.80) | 99.20 (0.98) | 92.80 (8.63) | 94.80 (4.12) | 94.40 (8.52) | 98.00 (2.19) | 92.40 (2.33) | 96.00 (1.79) |
| | 0.7 | 98.38 (2.16) | **99.46** (1.32) | 93.51 (2.16) | 95.14 (3.15) | 93.51 (7.37) | **99.46** (1.08) | 88.65 (10.59) | 94.00 (3.27) |
| | 0.8 | 98.40 (3.20) | **99.20** (1.96) | 96.80 (3.92) | 97.60 (3.20) | 86.40 (10.31) | **99.20** (1.60) | 96.80 (3.92) | 95.00 (4.47) |
| | 0.9 | **100.00** (0.00) | 98.33 (0.00) | 95.00 (6.67) | 96.67 (4.08) | 96.67 (6.67) | 98.33 (3.33) | 95.00 (4.08) | 96.00 (4.90) |
| PC | 0.1 | 87.84 (2.74) | 89.18 (1.89) | 86.98 (1.80) | 89.18 (1.53) | 85.05 (4.87) | 87.73 (2.12) | 84.41 (3.33) | **90.63** (0.84) |
| | 0.2 | 89.19 (1.31) | 88.49 (0.65) | 88.37 (1.60) | 87.79 (0.64) | 90.23 (2.42) | 90.12 (1.72) | 87.21 (3.83) | **92.12** (0.80) |
| | 0.3 | 91.26 (3.96) | **92.58** (2.22) | 88.61 (1.64) | 90.20 (1.84) | 92.45 (1.49) | 90.60 (1.84) | 88.58 (0.86) | 92.00 (1.03) |
| | 0.4 | 92.25 (3.40) | **94.11** (0.62) | 88.99 (2.57) | 90.08 (2.70) | 90.12 (3.85) | 92.71 (1.26) | 88.37 (1.34) | 93.50 (1.43) |
| | 0.5 | 93.33 (2.77) | **94.26** (1.59) | 89.35 (1.67) | 88.52 (0.45) | 92.13 (2.12) | 93.89 (1.39) | 87.41 (2.39) | 93.60 (1.20) |
| | 0.6 | 95.35 (1.64) | 94.65 (0.57) | 87.44 (1.14) | 89.30 (2.37) | 92.09 (2.69) | **95.64** (0.96) | 87.44 (3.07) | 95.00 (0.79) |
| | 0.7 | 95.94 (3.37) | 95.94 (2.92) | 88.44 (3.64) | 86.88 (2.12) | 92.81 (5.81) | 97.81 (0.77) | 85.63 (4.35) | **98.00** (1.94) |
| | 0.8 | 97.67 (2.08) | 97.67 (1.86) | 85.12 (3.15) | 86.63 (4.47) | 97.67 (2.94) | **99.42** (1.01) | 90.70 (3.60) | 98.50 (1.22) |
| | 0.9 | **100.00** (0.00) | **100.00** (0.00) | 90.48 (7.97) | 93.33 (7.13) | 98.10 (3.81) | **100.00** (0.00) | 91.43 (5.55) | 95.00 (0.00) |
| RD | 0.1 | 54.96 (4.41) | 56.99 (2.49) | 53.38 (3.49) | 53.19 (3.57) | **57.28** (4.13) | 54.91 (3.09) | 55.11 (1.90) | 56.70 (2.44) |
| | 0.2 | 58.33 (3.84) | **62.56** (2.82) | 56.94 (2.05) | 57.28 (2.22) | 57.56 (1.52) | 56.06 (1.27) | 59.22 (4.10) | 57.72 (3.03) |
| | 0.3 | 62.60 (5.90) | 62.79 (3.88) | 58.16 (0.79) | 58.35 (1.70) | 61.59 (4.07) | **66.79** (3.34) | 62.41 (1.04) | 58.97 (1.84) |
| | 0.4 | 67.33 (5.80) | **68.96** (2.57) | 59.41 (1.84) | 57.41 (1.72) | 62.96 (6.57) | 66.44 (2.91) | 65.33 (2.65) | 60.15 (1.63) |
| | 0.5 | 65.07 (5.91) | **77.60** (2.94) | 60.09 (2.60) | 58.84 (4.25) | 64.98 (4.66) | 70.76 (7.68) | 67.56 (1.97) | 61.45 (0.97) |
| | 0.6 | 74.78 (8.58) | 73.44 (5.87) | 60.11 (2.71) | 58.67 (3.74) | **75.11** (4.75) | 69.11 (4.83) | 70.67 (4.00) | 62.56 (2.47) |
| | 0.7 | **78.22** (7.85) | 77.48 (3.52) | 62.22 (4.11) | 62.67 (2.23) | 73.63 (9.31) | 75.11 (2.83) | 72.44 (4.67) | 63.54 (2.78) |
| | 0.8 | 74.44 (11.27) | **80.00** (6.00) | 60.22 (7.52) | 61.56 (3.41) | 78.89 (10.11) | 71.78 (10.27) | 73.11 (6.06) | 62.89 (3.69) |

| Dataset | Label Ratio | | | | | | | | |
|---|---|---|---|---|---|---|---|---|---|
| | 0.9 | 76.00 (12.36) | **81.33** (6.06) | 64.00 (5.14) | 63.11 (6.22) | 78.22 (8.60) | 80.44 (9.15) | 73.78 (6.65) | 70.50 (2.92) |
| SR | 0.1 | 96.47 (2.79) | 96.35 (1.45) | 92.99 (1.90) | 91.02 (2.34) | 94.31 (1.91) | 95.27 (1.01) | 94.31 (2.28) | **96.85** (0.85) |
| | 0.2 | 97.24 (1.76) | 96.43 (1.19) | 94.48 (1.32) | 94.95 (0.64) | 93.94 (1.13) | 95.89 (1.25) | 95.42 (1.22) | **97.93** (0.38) |
| | 0.3 | 98.69 (0.67) | 98.85 (0.57) | 94.77 (0.58) | 95.00 (1.00) | 92.92 (3.29) | 97.85 (0.52) | 94.62 (0.49) | **98.92** (0.15) |
| | 0.4 | 98.03 (1.05) | **99.37** (0.59) | 94.80 (0.54) | 95.34 (1.05) | 96.59 (3.27) | 98.21 (0.75) | 93.90 (2.73) | 98.64 (0.95) |
| | 0.5 | 98.82 (0.40) | 99.14 (0.55) | 96.02 (1.25) | 96.56 (1.21) | 97.74 (1.10) | 98.39 (0.90) | 96.34 (1.61) | **99.56** (0.65) |
| | 0.6 | 98.65 (1.21) | 98.78 (0.43) | 94.46 (2.82) | 95.00 (1.10) | 96.22 (5.26) | **99.59** (0.54) | 96.76 (1.44) | 99.29 (0.64) |
| | 0.7 | 98.56 (1.35) | **99.82** (0.67) | 95.14 (0.44) | 95.50 (1.27) | 98.50 (2.12) | 99.46 (0.72) | 97.48 (1.44) | 99.45 (0.45) |
| | 0.8 | 99.19 (1.62) | **99.73** (0.70) | 95.41 (2.20) | 94.05 (0.66) | 92.70 (9.92) | 99.19 (1.08) | 96.22 (1.01) | 99.43 (0.70) |
| | 0.9 | 98.92 (2.16) | **100.00** (0.00) | 96.76 (3.15) | 97.30 (1.71) | 96.22 (3.67) | **100.00** (0.00) | 98.38 (1.32) | 98.00 (1.63) |
| SL | 0.1 | 65.86 (1.71) | 69.36 (1.93) | 48.86 (8.58) | 49.06 (6.32) | 41.88 (2.61) | 58.42 (5.20) | 43.59 (3.75) | **74.33** (3.36) |
| | 0.2 | 76.70 (2.38) | **80.26** (2.66) | 67.52 (2.85) | 63.74 (3.46) | 45.11 (5.53) | 69.07 (5.16) | 46.15 (2.49) | 76.56 (3.83) |
| | 0.3 | 85.68 (2.63) | 85.17 (1.89) | 65.81 (3.69) | 65.25 (3.14) | 55.97 (11.67) | 79.92 (2.83) | 51.99 (5.56) | **85.96** (2.75) |
| | 0.4 | 87.26 (3.13) | **87.31** (1.77) | 67.70 (4.16) | 70.32 (2.18) | 68.84 (8.10) | 84.69 (2.98) | 54.96 (2.19) | 87.00 (1.44) |
| | 0.5 | 91.04 (3.30) | **91.69** (1.33) | 69.67 (2.43) | 70.21 (2.79) | 59.70 (11.92) | 91.16 (2.16) | 56.38 (2.69) | 88.36 (2.16) |
| | 0.6 | **93.41** (3.63) | 92.37 (1.03) | 75.04 (4.55) | 70.37 (3.63) | 59.26 (14.47) | 87.11 (2.02) | 58.30 (4.55) | 90.22 (2.18) |
| | 0.7 | **94.95** (4.49) | 94.26 (1.02) | 70.79 (1.21) | 74.16 (4.57) | 61.88 (15.15) | 87.13 (8.85) | 61.78 (6.11) | 94.80 (0.75) |
| | 0.8 | 94.96 (2.41) | **95.26** (1.44) | 73.78 (5.19) | 73.19 (4.74) | 63.11 (3.61) | 91.56 (1.37) | 62.22 (6.49) | 94.00 (1.32) |
| | 0.9 | 95.52 (3.78) | **97.01** (2.43) | 77.91 (2.39) | 71.04 (2.43) | 69.85 (8.57) | 87.46 (4.28) | 64.18 (10.21) | 94.33 (2.26) |
| SC | 0.1 | 89.69 (3.25) | 93.46 (1.21) | 93.95 (1.39) | 94.57 (1.23) | 93.15 (2.17) | 89.26 (3.94) | 90.68 (2.06) | **98.13** (1.14) |
| | 0.2 | 93.19 (3.76) | 97.08 (2.64) | 96.60 (1.41) | 97.50 (0.92) | 96.11 (0.94) | 94.93 (2.42) | 94.31 (0.84) | **98.79** (1.00) |
| | 0.3 | 96.11 (1.49) | 97.30 (1.56) | 96.51 (1.31) | 95.16 (1.87) | 96.35 (1.98) | 95.16 (1.80) | 95.56 (0.85) | **99.12** (0.16) |
| | 0.4 | 98.52 (0.90) | 97.78 (0.75) | 97.69 (0.59) | 97.13 (1.84) | 98.06 (1.15) | 96.39 (1.26) | 96.85 (0.90) | **99.33** (0.38) |
| | 0.5 | 99.11 (0.27) | 98.78 (0.57) | 97.56 (1.03) | 96.33 (1.67) | **99.22** (0.83) | 97.11 (0.65) | 97.22 (0.61) | 98.56 (1.14) |
| | 0.6 | **99.03** (0.56) | **99.03** (0.52) | 97.36 (2.54) | 97.08 (1.83) | 98.61 (0.88) | 98.89 (0.71) | 96.53 (2.28) | 98.29 (1.07) |
| | 0.7 | 99.44 (0.45) | 99.26 (0.00) | 98.33 (1.70) | 94.81 (4.17) | **99.63** (0.45) | 97.41 (1.23) | 96.67 (1.61) | 99.20 (0.40) |
| | 0.8 | 98.89 (1.04) | 98.61 (0.68) | 97.78 (0.68) | 96.67 (2.08) | **99.44** (0.68) | 99.17 (1.11) | 97.22 (1.76) | 99.14 (0.70) |
| | 0.9 | **100.00** (0.00) | **100.00** (0.00) | 98.89 (2.22) | 96.11 (2.22) | 99.44 (1.11) | 99.44 (1.11) | 97.22 (1.76) | 97.33 (2.49) |
| T1 | 0.1 | 69.86 (5.35) | 75.56 (4.74) | 73.75 (7.38) | 72.08 (4.55) | **76.94** (2.17) | 73.61 (4.85) | 75.69 (2.28) | 76.29 (3.08) |
| | 0.2 | 80.94 (5.54) | 86.25 (3.78) | 82.19 (2.00) | 84.38 (3.09) | 82.50 (4.70) | 79.84 (4.85) | 86.25 (4.88) | **88.50** (3.31) |
| | 0.3 | 86.25 (3.27) | **89.11** (1.87) | 84.64 (6.93) | 82.32 (4.87) | 80.89 (9.13) | 85.00 (4.32) | 86.07 (3.22) | 88.91 (2.78) |
| | 0.4 | 89.58 (2.95) | 90.42 (1.47) | 88.54 (5.51) | 85.83 (5.21) | **91.04** (2.76) | 90.83 (3.19) | 89.79 (2.41) | 86.00 (1.33) |
| | 0.5 | 93.50 (5.88) | **95.50** (3.59) | 91.50 (1.66) | 92.00 (1.70) | 93.75 (3.71) | 93.00 (4.72) | 92.25 (2.00) | 87.25 (3.30) |
| | 0.6 | **96.25** (3.64) | 94.38 (1.59) | 93.75 (3.28) | 87.81 (5.17) | 92.19 (3.70) | 93.44 (3.75) | 93.13 (4.59) | 90.33 (3.86) |
| | 0.7 | **97.50** (2.04) | 96.25 (1.56) | 93.75 (1.86) | 91.67 (2.64) | 95.42 (4.25) | 93.33 (4.04) | 92.92 (5.53) | 91.50 (2.00) |
| | 0.8 | **98.13** (3.75) | 96.88 (1.98) | 92.50 (3.75) | 92.50 (4.68) | 97.50 (3.64) | 95.00 (3.75) | 95.00 (2.50) | 91.33 (5.81) |
| | 0.9 | 95.00 (7.29) | 97.50 (0.00) | 90.00 (5.00) | 96.25 (5.00) | 96.25 (3.06) | 97.50 (3.06) | **98.75** (2.50) | 92.00 (4.00) |
| TR | 0.1 | 75.74 (9.23) | 76.30 (13.61) | 75.00 (2.11) | 82.78 (10.4) | 74.63 (4.74) | 72.59 (5.7) | 83.15 (7.09) | **87.20** (10.65) |
| | 0.2 | 90.00 (2.43) | 93.75 (3.81) | 84.58 (6.89) | 91.04 (5.21) | 83.54 (8.24) | 90.21 (4.91) | 95.00 (4.14) | **99.11** (9.92) |
| | 0.3 | 94.76 (4.91) | 96.43 (2.71) | 93.33 (9.72) | 91.90 (10.03) | 92.14 (3.50) | 95.95 (4.36) | 94.52 (3.42) | **98.75** (2.67) |
| | 0.4 | 96.39 (1.42) | 98.33 (2.58) | 93.33 (8.02) | 93.89 (6.37) | 94.44 (0.88) | 91.67 (7.81) | 98.61 (1.24) | **99.43** (1.40) |
| | 0.5 | 97.67 (2.26) | 99.00 (1.25) | 92.33 (2.26) | 96.67 (3.80) | 93.67 (5.81) | 96.33 (4.00) | 97.33 (0.82) | **100.00** (0.00) |
| | 0.6 | 97.92 (2.28) | 99.17 (2.64) | 97.08 (2.12) | 97.08 (2.83) | 95.00 (7.05) | 96.67 (4.68) | **99.58** (0.83) | 99.50 (1.00) |
| | 0.7 | 96.11 (2.22) | 98.89 (1.36) | 96.67 (3.24) | 95.56 (4.51) | **99.44** (1.11) | 96.67 (5.39) | 96.67 (2.72) | 99.33 (0.00) |
| | 0.8 | 99.17 (1.67) | **100.00** (0.00) | 92.50 (3.12) | 95.00 (3.12) | 98.33 (2.04) | 97.50 (2.04) | **100.00** (0.00) | **100.00** (0.00) |
| | 0.9 | **100.00** (0.00) | **100.00** (0.00) | 98.33 (3.33) | 98.33 (3.33) | 98.33 (3.33) | **100.00** (0.00) | **100.00** (0.00) | **100.00** (0.00) |
| TP | 0.1 | 99.18 (0.30) | 99.29 (0.19) | 98.69 (0.19) | 98.90 (0.12) | 97.24 (1.58) | 99.07 (0.61) | 93.02 (3.56) | **99.95** (0.06) |
| | 0.2 | 99.81 (0.10) | 99.67 (0.11) | 99.04 (0.37) | 99.37 (0.23) | 98.78 (0.26) | 99.70 (0.11) | 94.97 (1.06) | **99.95** (0.05) |
| | 0.3 | 99.91 (0.11) | 99.85 (0.12) | 99.06 (0.29) | 99.42 (0.11) | 99.04 (1.38) | 99.59 (0.38) | 96.67 (0.94) | **99.98** (0.02) |
| | 0.4 | 99.90 (0.09) | 99.93 (0.11) | 99.51 (0.32) | 99.33 (0.35) | 98.58 (1.17) | 99.84 (0.10) | 96.53 (0.70) | **99.99** (0.02) |
| | 0.5 | 99.95 (0.05) | 99.84 (0.08) | 99.35 (0.25) | 99.53 (0.38) | 99.79 (0.13) | 99.88 (0.03) | 97.80 (0.73) | **99.99** (0.03) |
| | 0.6 | 99.93 (0.06) | 99.87 (0.03) | 99.17 (0.19) | 99.52 (0.12) | 98.78 (1.38) | 99.92 (0.09) | 97.62 (0.28) | **99.97** (0.04) |
| | 0.7 | 99.87 (0.13) | 99.98 (0.05) | 99.44 (0.25) | 99.33 (0.55) | 99.31 (0.71) | 99.87 (0.08) | 98.13 (0.79) | **100.00** (0.00) |
| | 0.8 | 99.87 (0.27) | 99.97 (0.08) | 99.33 (0.62) | 99.70 (0.24) | 99.80 (0.24) | 99.90 (0.13) | 97.90 (1.02) | **99.97** (0.07) |
| | 0.9 | 99.93 (0.13) | **100.00** (0.00) | 99.40 (0.39) | 99.47 (0.50) | 99.73 (0.25) | 99.93 (0.13) | 98.53 (0.62) | **100.00** (0.00) |
| YO | 0.1 | 74.99 (1.21) | 74.30 (1.76) | 61.57 (1.15) | 61.36 (2.53) | 68.46 (2.52) | 74.23 (1.50) | 69.69 (2.90) | **75.12** (4.66) |
| | 0.2 | 81.10 (2.50) | **81.70** (1.49) | 61.78 (0.97) | 62.73 (3.17) | 74.07 (1.95) | 80.24 (1.81) | 70.90 (2.58) | 79.97 (5.05) |
| | 0.3 | **86.06** (2.17) | 86.00 (1.51) | 65.45 (2.30) | 61.27 (1.04) | 74.99 (5.06) | 83.80 (1.66) | 75.17 (1.71) | 83.43 (7.23) |
| | 0.4 | 87.98 (1.81) | 87.59 (1.78) | 65.39 (4.77) | 61.65 (0.74) | 80.61 (2.61) | 84.24 (3.15) | 72.41 (1.60) | **88.90** (5.98) |
| | 0.5 | 90.59 (2.45) | 89.52 (1.41) | 65.07 (1.92) | 65.49 (1.35) | 82.83 (1.86) | 85.07 (5.00) | 78.65 (1.82) | **92.65** (0.73) |
| | 0.6 | **92.53** (3.63) | 91.39 (0.87) | 67.20 (5.98) | 61.44 (2.19) | 80.51 (5.89) | 84.27 (4.89) | 80.30 (1.76) | 90.66 (5.18) |
| | 0.7 | 91.99 (2.93) | **93.06** (2.48) | 65.32 (5.60) | 66.70 (2.72) | 81.18 (3.67) | 89.26 (2.79) | 80.34 (2.10) | 90.58 (8.36) |
| | 0.8 | 94.60 (3.08) | 94.29 (0.87) | 65.20 (2.42) | 66.46 (3.39) | 79.04 (9.23) | 88.69 (1.73) | 81.46 (4.09) | **95.49** (1.77) |
| | 0.9 | **94.95** (2.02) | 93.23 (2.40) | 65.96 (4.01) | 68.38 (5.25) | 82.93 (2.60) | 90.00 (3.86) | 84.24 (3.05) | 91.47 (3.99) |

Table 9: Classification performance of ablation models and MDTA for various label ratios $\in [0.1, 0.9]$. The value in parentheses denotes the standard deviation. For each label ratio in datasets, the best score is highlighted in boldface.

| Dataset | Label Ratio | MDTA (ours) | MDTA w/o $f_1$ | MDTA w/o $\mathcal{L}_{RP}$ | MDTA w/o RM |
|---|---|---|---|---|---|
| | 0.1 | **99.78** (0.26) | 99.46 (0.65) | 99.46 (0.43) | 98.39 (0.71) |
| | 0.2 | 99.57 (0.40) | 99.64 (0.26) | 98.57 (0.87) | **99.82** (0.22) |
| | 0.3 | **99.68** (0.43) | 99.64 (0.43) | 98.75 (0.80) | 99.10 (0.68) |
| | 0.4 | **99.78** (0.26) | 99.64 (0.43) | 99.10 (0.43) | 98.92 (0.79) |
| CB | 0.5 | 99.57 (0.40) | **99.64** (0.26) | 98.57 (0.73) | 99.10 (0.68) |
| | 0.6 | **99.57** (0.22) | 99.28 (0.53) | 99.46 (0.26) | 99.46 (0.43) |
| | 0.7 | 99.78 (0.43) | **99.82** (0.22) | 99.28 (0.63) | 99.28 (0.40) |
| | 0.8 | **100.00** (0.00) | 99.64 (0.43) | 99.10 (0.48) | 99.28 (0.40) |
| | 0.9 | **99.68** (0.43) | 99.64 (0.26) | 99.46 (0.43) | 99.64 (0.26) |

| | | | | | |
|---|---|---|---|---|---|
| CX | 0.1 | **49.10** (4.82) | 22.65 (2.79) | 28.42 (3.52) | 41.88 (3.95) |
| | 0.2 | 48.97 (3.96) | 24.36 (5.48) | 50.85 (3.52) | **52.35** (1.55) |
| | 0.3 | **61.92** (3.68) | 31.62 (4.01) | 55.77 (2.21) | 52.99 (5.36) |
| | 0.4 | **65.64** (3.88) | 34.62 (4.28) | 62.61 (2.53) | 60.04 (4.31) |
| | 0.5 | **67.56** (2.35) | 32.91 (4.00) | 65.17 (2.34) | 57.26 (9.82) |
| | 0.6 | **68.97** (4.67) | 39.53 (10.11) | 64.32 (1.94) | 68.16 (3.11) |
| | 0.7 | **70.26** (4.01) | 39.32 (4.32) | 62.82 (7.64) | 67.95 (2.61) |
| | 0.8 | 71.41 (3.33) | 40.17 (3.38) | 69.02 (3.55) | **74.15** (2.54) |
| | 0.9 | 72.82 (1.32) | 42.95 (9.15) | 70.09 (2.70) | **77.14** (1.04) |
| EF | 0.1 | 99.55 (0.42) | 78.91 (2.35) | **100.00** (0.00) | 99.06 (0.42) |
| | 0.2 | **99.89** (0.23) | 81.73 (2.03) | 99.62 (0.45) | 99.81 (0.23) |
| | 0.3 | **99.89** (0.23) | 81.36 (1.37) | 99.81 (0.23) | 99.81 (0.23) |
| | 0.4 | 99.66 (0.45) | 81.73 (0.42) | **100.00** (0.00) | 99.25 (0.66) |
| | 0.5 | 99.77 (0.45) | 82.30 (1.31) | **99.81** (0.23) | 99.06 (0.71) |
| | 0.6 | **99.89** (0.23) | 82.86 (3.45) | 99.62 (0.45) | 99.81 (0.23) |
| | 0.7 | 99.66 (0.45) | 82.49 (5.03) | **100.00** (0.00) | 99.44 (0.45) |
| | 0.8 | **100.00** (0.00) | 88.70 (3.73) | **100.00** (0.00) | 99.81 (0.23) |
| | 0.9 | **100.00** (0.00) | 82.30 (3.18) | **100.00** (0.00) | **100.00** (0.00) |
| L2 | 0.1 | **61.60** (4.08) | 52.00 (7.33) | 52.00 (7.42) | 61.33 (5.99) |
| | 0.2 | **73.60** (4.08) | 60.00 (1.60) | 54.67 (7.33) | 57.33 (13.72) |
| | 0.3 | 66.40 (5.99) | 57.33 (8.16) | **69.33** (6.50) | 57.33 (12.80) |
| | 0.4 | 67.20 (13.00) | 64.00 (7.76) | **80.00** (4.08) | 54.67 (15.05) |
| | 0.5 | 76.80 (6.88) | 76.00 (1.60) | **85.33** (1.96) | 74.67 (5.43) |
| | 0.6 | **85.60** (3.20) | 78.67 (2.53) | 78.67 (5.99) | 81.33 (5.31) |
| | 0.7 | **84.30** (2.53) | 76.00 (3.20) | 78.67 (5.43) | 73.33 (7.42) |
| | 0.8 | **78.40** (5.99) | 74.67 (2.99) | 76.00 (6.88) | 72.00 (3.92) |
| | 0.9 | **83.20** (6.88) | 72.00 (3.20) | 78.67 (2.99) | 74.67 (4.08) |
| MS | 0.1 | **91.14** (0.95) | 81.96 (3.27) | 88.76 (1.51) | 90.07 (1.39) |
| | 0.2 | **95.29** (0.43) | 90.98 (0.46) | 92.68 (1.20) | 93.59 (1.41) |
| | 0.3 | **95.22** (1.00) | 90.59 (1.37) | 91.11 (2.87) | 93.59 (1.38) |
| | 0.4 | 95.14 (1.60) | 91.90 (0.97) | 94.64 (0.91) | **96.34** (0.46) |
| | 0.5 | **95.06** (1.01) | 88.10 (2.26) | 93.07 (0.64) | 93.99 (0.74) |
| | 0.6 | 95.37 (1.56) | 90.59 (2.18) | 93.73 (1.00) | **95.56** (0.77) |
| | 0.7 | 96.24 (0.64) | 92.16 (1.42) | 94.12 (0.91) | **96.60** (0.46) |
| | 0.8 | **95.69** (0.61) | 93.46 (1.13) | 93.46 (1.30) | 95.16 (1.01) |
| | 0.9 | **96.08** (0.61) | 93.73 (1.04) | 95.69 (0.46) | 95.82 (0.72) |
| PL | 0.1 | 95.71 (0.95) | 93.65 (1.17) | 96.83 (0.95) | **97.62** (0.95) |
| | 0.2 | 95.71 (1.78) | 94.44 (1.78) | **96.83** (1.51) | **96.83** (1.78) |
| | 0.3 | 95.71 (1.78) | 93.65 (2.13) | **96.03** (1.74) | 95.24 (0.95) |
| | 0.4 | 95.71 (1.78) | 93.65 (2.33) | **97.62** (1.90) | 96.03 (1.78) |
| | 0.5 | 96.67 (1.90) | 93.65 (1.17) | **97.62** (1.90) | 93.65 (4.10) |
| | 0.6 | **97.62** (1.51) | 93.65 (1.90) | 96.03 (3.01) | **97.62** (0.95) |
| | 0.7 | 96.67 (2.43) | 94.44 (1.51) | **97.62** (1.90) | 95.24 (2.43) |
| | 0.8 | **99.05** (1.17) | 95.24 (1.90) | 98.41 (1.17) | 96.83 (1.51) |
| | 0.9 | **99.05** (1.90) | 92.06 (2.43) | 98.41 (1.17) | 96.03 (3.01) |
| PC | 0.1 | **86.67** (2.26) | 85.65 (0.88) | 85.65 (1.67) | 85.19 (1.84) |
| | 0.2 | **91.67** (1.76) | 86.11 (0.68) | 88.43 (3.97) | 91.20 (2.42) |
| | 0.3 | 93.33 (1.62) | 87.04 (1.88) | **94.91** (1.36) | 89.81 (2.42) |
| | 0.4 | 90.56 (1.62) | 85.65 (2.48) | 89.81 (1.42) | **91.67** (1.42) |
| | 0.5 | **94.44** (2.91) | 88.89 (1.42) | 92.59 (2.83) | 89.81 (2.22) |
| | 0.6 | **94.44** (1.24) | 87.04 (2.69) | 93.98 (2.39) | 93.98 (2.39) |
| | 0.7 | **95.56** (1.84) | 87.50 (1.67) | 95.37 (1.36) | 95.37 (1.88) |
| | 0.8 | **97.22** (0.88) | 87.50 (2.91) | 95.83 (1.88) | 96.76 (1.36) |
| | 0.9 | **98.33** (0.56) | 85.65 (0.56) | 97.69 (0.68) | 96.30 (1.04) |
| RD | 0.1 | 53.60 (2.09) | 48.89 (1.24) | **54.00** (2.41) | 52.00 (2.17) |
| | 0.2 | 58.00 (3.65) | 54.00 (2.78) | **58.89** (4.11) | 54.89 (3.44) |
| | 0.3 | 57.07 (2.29) | 52.00 (3.17) | **58.44** (4.17) | 53.11 (4.69) |
| | 0.4 | 58.27 (3.20) | 57.78 (3.56) | **58.67** (3.02) | 57.78 (4.16) |
| | 0.5 | **61.07** (1.16) | 50.22 (4.59) | 60.00 (2.25) | 59.78 (1.42) |
| | 0.6 | 60.40 (3.64) | 56.00 (2.52) | **62.44** (1.19) | 61.11 (2.60) |
| | 0.7 | 61.73 (3.26) | 54.44 (4.37) | **62.67** (3.06) | 59.33 (2.60) |
| | 0.8 | 62.27 (4.10) | 52.22 (4.64) | 61.56 (2.25) | **62.44** (2.51) |
| | 0.9 | 63.60 (2.82) | 59.11 (1.95) | **64.00** (1.71) | 63.56 (3.14) |
| SR | 0.1 | **99.52** (0.39) | 92.80 (2.12) | 98.67 (0.60) | 99.47 (0.64) |
| | 0.2 | 99.36 (0.60) | 93.33 (2.30) | **99.47** (0.39) | 99.20 (0.64) |
| | 0.3 | 99.04 (0.93) | 94.13 (1.09) | 98.67 (0.72) | **99.47** (0.39) |
| | 0.4 | 99.52 (0.64) | 93.87 (1.18) | **99.73** (0.32) | 99.47 (0.39) |
| | 0.5 | 99.84 (0.32) | 92.80 (2.02) | **100.00** (0.00) | 99.20 (0.32) |
| | 0.6 | **99.52** (0.64) | 93.87 (1.68) | 99.47 (0.39) | 99.20 (0.96) |
| | 0.7 | **100.00** (0.00) | 94.40 (1.87) | **100.00** (0.00) | 99.20 (0.39) |
| | 0.8 | **99.84** (0.32) | 94.40 (0.93) | 99.20 (0.39) | 99.47 (0.64) |
| | 0.9 | **99.84** (0.32) | 94.13 (1.72) | 99.20 (0.64) | 99.47 (0.64) |
| SL | 0.1 | **74.67** (3.80) | 56.15 (2.09) | 54.52 (4.39) | 67.70 (6.53) |
| | 0.2 | **76.62** (3.43) | 60.30 (2.15) | 76.59 (2.29) | 76.15 (4.20) |
| | 0.3 | 85.07 (3.09) | 63.85 (2.70) | **85.33** (1.24) | 81.63 (2.95) |
| | 0.4 | **87.73** (1.55) | 67.56 (4.26) | 86.81 (2.09) | 81.48 (1.72) |

| | | | | | |
|---|---|---|---|---|---|
| | 0.5 | 87.29 (1.24) | 68.15 (3.72) | **89.19** (1.55) | 85.33 (2.46) |
| | 0.6 | **89.96** (1.79) | 70.37 (2.82) | 88.59 (1.74) | 89.04 (1.34) |
| | 0.7 | **92.00** (1.57) | 77.48 (6.07) | 90.52 (1.76) | 88.44 (1.07) |
| | 0.8 | **90.84** (2.22) | 76.89 (4.50) | 89.48 (1.32) | 89.78 (0.82) |
| | 0.9 | **91.47** (1.79) | 79.26 (4.52) | 91.41 (0.28) | 90.67 (1.62) |
| | 0.1 | 96.67 (2.11) | 60.56 (6.74) | 94.44 (1.05) | **96.94** (0.91) |
| | 0.2 | 97.00 (1.35) | 77.50 (4.10) | 97.50 (0.85) | **98.06** (0.53) |
| | 0.3 | **99.00** (0.62) | 81.94 (4.28) | 98.06 (0.82) | 97.22 (1.13) |
| | 0.4 | **98.67** (0.85) | 86.94 (3.00) | 96.11 (1.25) | 97.78 (1.11) |
| SC | 0.5 | **98.00** (0.85) | 87.50 (3.70) | 96.11 (1.55) | 97.78 (1.11) |
| | 0.6 | 97.67 (1.11) | 88.61 (3.03) | 97.50 (1.18) | **98.61** (0.62) |
| | 0.7 | **98.50** (0.97) | 88.06 (4.20) | 98.06 (0.53) | 98.06 (1.13) |
| | 0.8 | 98.17 (1.22) | 92.22 (3.27) | 95.56 (1.87) | **99.17** (0.33) |
| | 0.9 | **99.33** (0.62) | 91.39 (3.59) | 98.33 (0.82) | 98.33 (0.67) |
| | 0.1 | **82.59** (1.89) | 76.54 (5.81) | 79.63 (4.60) | 76.54 (2.77) |
| | 0.2 | 89.26 (4.12) | 86.42 (5.62) | **90.74** (1.48) | 89.51 (2.72) |
| | 0.3 | **96.67** (2.16) | 88.89 (3.19) | 89.51 (5.16) | 93.21 (3.01) |
| | 0.4 | 95.56 (2.22) | 89.51 (6.35) | 91.98 (2.77) | **96.30** (1.48) |
| T1 | 0.5 | **94.81** (2.96) | 93.21 (2.34) | 92.59 (3.31) | 92.59 (2.46) |
| | 0.6 | **97.04** (1.48) | 95.68 (1.17) | 95.06 (2.03) | 95.06 (1.66) |
| | 0.7 | **97.04** (2.77) | 95.68 (1.39) | 95.06 (0.91) | 92.59 (2.72) |
| | 0.8 | **96.30** (3.10) | 96.30 (0.00) | 95.68 (0.74) | 93.83 (3.59) |
| | 0.9 | 97.04 (1.48) | 94.44 (0.74) | **97.53** (1.39) | **97.53** (1.39) |
| | 0.1 | **90.00** (10.25) | 75.00 (7.91) | **90.00** (9.00) | 79.17 (9.80) |
| | 0.2 | 99.50 (1.00) | 91.67 (3.00) | **100.00** (0.00) | 88.33 (8.94) |
| | 0.3 | **99.00** (1.22) | 96.67 (1.87) | 98.33 (1.22) | 97.50 (1.22) |
| | 0.4 | 99.50 (1.00) | 97.50 (1.00) | **100.00** (0.00) | 98.33 (2.00) |
| TR | 0.5 | **100.00** (0.00) | 97.50 (3.00) | **100.00** (0.00) | **100.00** (0.00) |
| | 0.6 | **100.00** (0.00) | 99.17 (1.00) | **100.00** (0.00) | **100.00** (0.00) |
| | 0.7 | 99.00 (2.00) | 98.33 (1.22) | **100.00** (0.00) | **100.00** (0.00) |
| | 0.8 | **100.00** (0.00) | 99.17 (1.00) | 99.17 (1.00) | 98.33 (2.00) |
| | 0.9 | 99.00 (1.22) | 95.83 (2.24) | 99.17 (1.00) | **100.00** (0.00) |
| | 0.1 | **99.94** (0.08) | 95.17 (1.68) | 99.67 (0.15) | 99.80 (0.19) |
| | 0.2 | 99.94 (0.08) | 93.67 (5.53) | 99.83 (0.20) | **100.00** (0.00) |
| | 0.3 | **100.00** (0.00) | 97.90 (1.10) | 99.83 (0.09) | **100.00** (0.00) |
| | 0.4 | **100.00** (0.00) | 99.47 (0.21) | 99.97 (0.04) | 99.90 (0.12) |
| TP | 0.5 | 99.96 (0.08) | 98.30 (0.64) | 99.80 (0.19) | **99.97** (0.04) |
| | 0.6 | 99.96 (0.05) | 99.50 (0.23) | **99.97** (0.04) | 99.90 (0.05) |
| | 0.7 | **100.00** (0.00) | 99.67 (0.19) | 99.93 (0.05) | 99.80 (0.14) |
| | 0.8 | 99.96 (0.05) | 99.73 (0.09) | 99.90 (0.12) | **100.00** (0.00) |
| | 0.9 | **100.00** (0.00) | 99.57 (0.23) | **100.00** (0.00) | **100.00** (0.00) |
| | 0.1 | **73.48** (5.14) | 65.91 (3.49) | 71.46 (4.15) | 71.16 (3.89) |
| | 0.2 | **78.45** (5.09) | 72.32 (3.20) | 71.82 (6.39) | 76.31 (4.33) |
| | 0.3 | 81.73 (6.28) | 74.14 (1.96) | 80.96 (2.07) | **87.78** (0.46) |
| | 0.4 | 86.03 (5.85) | 74.29 (4.98) | 76.67 (5.89) | **87.02** (2.11) |
| YO | 0.5 | 89.36 (0.65) | 79.09 (4.38) | 77.02 (5.99) | **90.91** (0.69) |
| | 0.6 | **88.03** (4.09) | 79.29 (2.57) | 79.55 (1.69) | 80.15 (5.63) |
| | 0.7 | **88.33** (7.22) | 83.64 (1.46) | 83.84 (3.26) | 87.68 (2.52) |
| | 0.8 | **92.27** (2.88) | 84.24 (1.99) | 83.54 (3.52) | 86.16 (5.02) |
| | 0.9 | 88.82 (4.23) | 83.08 (2.71) | 79.80 (5.31) | **90.25** (2.34) |

