# OpenReview forum: "Masked Dual-Temporal Autoencoders for Semi-Supervised Time-Series Classification"
_ICLR.cc/2024/Conference — Submitted to ICLR 2024_

### Official Review · Reviewer_q17s · 2023-10-23

**Soundness:** 2 fair
**Presentation:** 3 good
**Contribution:** 2 fair
**Rating:** 6
**Confidence:** 2

**Summary:**

This paper introduces the idea of Masked Autoencoders into semi-supervised time-series classification. The proposed method, MDTA,  effectively captures semantic information of time series by reflecting diverse temporal resolutions without temporal information loss and the authors further incorporate the extracted semantic information to enhance classification performance. Extensive evaluation on multiple datasets validates the effectiveness of this method.

**Strengths:**

1. The evaluation is extensive. The proposed method is empirically compared with recent competing approaches and the authors perform systematic ablation studies to validate each aspect of the proposed method's design.
2. Results seem promising. MDTA exhibits leading performance on multiple datasets compared with existing works.
3. Writing is clear. The presentation is clear and the method is easy to understand.

**Weaknesses:**

1. Training/inference efficiency. Although MDTA obtains better performance compared with baseline methods, it still remains unclear whether it will cost higher training or inference costs. For example, the authors could compare the training time, number of parameters, GPU latency between MDTA and baseline methods.
2. Results on transductive inferences. It seems that the performance of MDTA in Tab. 6 is not as good as those in Tab. 5, the authors are suggested to summarize the average performance in another table and what are the possible reasons behind this.
3. Pretrain and fine-tuning results. One advantage of Masked Autoencoders is that the model can learn very general representations and the pre-trained model can be easily applied to other datasets or tasks with quick fine-tuning operations. The reviewer wonders whether MDTA can also be utilized for this strategy.

**Questions:**

1. Comparison with more recent works. The reviewer notices that the most recent baselines used in the paper were published in 2022. Are there any other recent works of semi-supervised time-series classification?

---

> ### Author Response · Authors · 2023-11-17
> **Response for Reviewer q17s by Authors**
>
> We appreciate your valuable comments. We tried to address all your comments, including three weaknesses, W1-W3, and a question, Q1. We hope our responses help to clarify our work.
>
> **W1** We provide the training time per epoch of the proposed method with that of baseline methods in the table below. As mentioned in Section 5 of the manuscript, a limitation of MDTA is relatively low efficiency compared to the other baselines because it employs two consecutive sub-encoders.
>
> Although we used the transformer architecture introduced in [1], the first proposed transformer for time series to extract their useful representations, its computational inefficiency has also been demonstrated in [2]. In other words, the efficiency of our approach can be improved by replacing the transformer architecture with one of the recent transformers, improving computational efficiency. In addition, another possible solution for future research direction is to devise a lightweight transformer encoder architecture that can reflect diverse temporal resolutions.
>
> We added this limitation and future research direction in Section 5 of the revised manuscript and provided the results of comparing training time in the appendix as below.
>
> | *Training time (sec)* | Pseudo | $\Pi$-model | FixMatch |  MTL | SSTSC | iTimes |  MDTA |
> |:---------------------:|:------:|:------:|:--------:|:----:|:-----:|:------:|:-----:|
> |          CBF          |  0.57  |  0.97  |   0.98   | 0.90 |  1.39 |  0.93  |  5.74 |
> |        CricketX       |  0.48  |  0.83  |   0.85   | 0.79 |  1.19 |  0.80  |  4.83 |
> |      ECGFiveDays      |  0.54  |  0.92  |   0.95   | 0.89 |  1.32 |  0.90  |  5.45 |
> |       Lightning2      |  0.09  |  0.14  |   0.15   | 0.14 |  0.21 |  0.14  |  0.79 |
> |       MoteStrain      |  0.76  |  1.29  |   1.34   | 1.23 |  1.90 |  1.30  |  7.88 |
> |         Plane         |  0.13  |  0.22  |   0.23   | 0.21 |  0.31 |  0.21  |  1.25 |
> |       PowerCons       |  0.23  |  0.39  |   0.40   | 0.36 |  0.55 |  0.37  |  2.20 |
> |  RefrigerationDevices |  0.48  |  0.85  |   0.88   | 0.80 |  1.19 |  0.83  |  5.01 |
> | SonyAIBORobotSurface1 |  0.38  |  0.66  |   0.68   | 0.60 |  0.93 |  0.63  |  3.80 |
> |      SwedishLeaf      |  0.71  |  1.21  |   1.24   | 1.13 |  1.73 |  1.16  |  6.95 |
> |    SyntheticControl   |  0.37  |  0.63  |   0.65   | 0.62 |  0.92 |  0.61  |  3.74 |
> |    ToeSegmentation1   |  0.17  |  0.29  |   0.30   | 0.28 |  0.42 |  0.28  |  1.65 |
> |         Trace         |  0.13  |  0.22  |   0.22   | 0.21 |  0.31 |  0.21  |  1.23 |
> |      TwoPatterns      |  3.05  |  5.06  |   5.68   | 4.81 |  7.77 |  4.86  | 31.94 |
> |          Yoga         |  1.91  |  3.47  |   3.58   | 3.26 |  4.85 |  3.35  | 21.55 |
>
> [1] Zerveas, G., Jayaraman, S., Patel, D., Bhamidipaty, A., & Eickhoff, C. (2021, August). A transformer-based framework for multivariate time series representation learning. In Proceedings of the 27th ACM SIGKDD conference on knowledge discovery & data mining (pp. 2114-2124).
>
> [2] Cheng, M., Liu, Q., Liu, Z., Li, Z., Luo, Y., & Chen, E. (2023). FormerTime: Hierarchical Multi-Scale Representations for Multivariate Time Series Classification. arXiv preprint arXiv:2302.09818.

---

> ### Author Response · Authors · 2023-11-17
> **Response for Reviewer q17s by Authors (cont'd)**
>
> **W2** The proposed method and all baselines are semi-supervised learning approaches mainly for inductive inference, which involves measuring model performance using a test dataset separate from the training dataset. Thus, for the inductive inference, our method achieves notable performance improvement.
>
> By contrast, transductive inference evaluates model performance for unlabeled instances in the training dataset. Since MDTA focuses on enhancing the model's generalization performance on the test datasets, it may not well-perform on transductive inference. Nevertheless, as shown in the table below, MDTA and Pseudo perform better than the others under the transductive setting. Although the performance of the two methods is comparable, MDTA performs remarkably better than Pseudo in some datasets, such as Lightning2, Trace, and TwoPatterns.
>
> |    Dataset    | CE | Pseudo | $\Pi$-model | FixMatch |  MTL  | SSTSC | iTimes |  MDTA |
> |:---------:|:-----:|:------:|:------:|:--------:|:-----:|:-----:|:------:|:-----:|
> |          CBF          | 99.29 |  99.56 |  99.21 |   99.27  | 98.63 | 99.59 |  99.42 | **99.65** |
> |        CricketX       | 69.65 |  69.76 |  65.57 |   63.74  | 55.97 | 58.49 |  **75.53** | 68.87 |
> |      ECGFiveDays      | 99.20 |  99.34 |  87.56 |   87.76  | 99.25 | 98.73 |  96.46 | **99.78** |
> |       Lightning2      | 76.77 |  73.52 |  66.51 |   68.57  | 77.28 | 75.60 |  75.44 | **79.02** |
> |       MoteStrain      | **95.84** |  95.77 |  94.80 |   94.54  | 93.13 | 95.20 |  94.78 | 95.38 |
> |         Plane         | 97.72 |  **97.79** |  85.58 |   88.43  | 87.85 | 96.33 |  86.26 | 96.02 |
> |       PowerCons       | 93.65 |  94.10 |  88.20 |   89.10  | 92.30 | 94.21 |  87.91 | **94.26** |
> |  RefrigerationDevices | 67.97 |  **71.24** |  59.39 |   59.01  | 67.80 | 67.93 |  66.63 | 61.61 |
> | SonyAIBORobotSurface1 | **98.28** |  98.72 |  94.98 |   94.97  | 95.46 | 98.20 |  95.94 | 98.67 |
> |      SwedishLeaf      | 87.27 |  **88.08** |  68.56 |   67.48  | 58.40 | 81.83 |  55.51 | 87.28 |
> |    SyntheticControl   | 97.11 |  97.92 |  97.18 |   96.15  | 97.78 | 96.42 |  95.81 | **98.65** |
> |    ToeSegmentation1   | 89.67 |  **91.31** |  87.85 |   87.20  | 89.61 | 89.06 |  89.98 | 88.01 |
> |         Trace         | 94.19 |  95.76 |  91.46 |   93.58  | 92.17 | 93.07 |  96.10 | **98.15** |
> |      TwoPatterns      | 99.82 |  99.82 |  99.22 |   99.40  | 99.01 | 99.75 |  96.80 | **99.98** |
> |          Yoga         | **88.31** |  87.90 |  64.77 |   63.94  | 78.29 | 84.42 |  77.02 | 87.59 |
> |      *Average rank*     |  3.07 |  2.13  |  6.60  |   6.60   |  5.53 |  4.20 |  5.40  |  2.47 |
>
> We added this analysis in the appendix.
>
> **W3** As the reviewer mentioned, one advantage of masked autoencoders is to learn high-quality representations. In this work, we used this advantage to extract valuable semantic information from unlabeled time-series instances for effectively performing semi-supervised time-series classification. In other words, we enhanced classification performance under label sparsity by incorporating the semantic information obtained from unlabeled instances with supervisory features, including hard-to-learn class information, learned from labeled ones.
>
> It is also possible to first learn universal representations by masked time-series modeling and then fine-tune them to be suitable for time-series classification. However, this two-stage approach can show inevitable performance gaps in time-series classification because they are learned by focusing on being generalizable for various tasks, not jointly learned with the label information [1]. Thus, if we aim to perform specific supervised tasks, jointly using label information and masked time-series modeling is likely to be effective.
>
> [1] Qi, G. J., & Luo, J. (2020). Small data challenges in big data era: A survey of recent progress on unsupervised and semi-supervised methods. IEEE Transactions on Pattern Analysis and Machine Intelligence, 44(4), 2168-2187.

---

> ### Author Response · Authors · 2023-11-17
> **Response for Reviewer q17s by Authors (cont'd)**
>
> **Q1** We further compared the proposed method with a recent work, CA-TCC [1], published in August 2023. The table below shows the average accuracy scores across label ratios from 0.1 to 0.9 under inductive inference. Here, we present the name of each dataset as an abbreviation. The proposed method performs better than CA-TCC on average, achieving better performance in 10 out of 15 datasets. Especially for several datasets, such as PowerCons (PC), SonyAIBORobotSurface1 (SR), SwedishLeaf (SL), SyntheticControl (SC), ToeSegmentation1 (T1), Trace (TR), and Yoga (YO), our method shows overwhelming performance compared to CA-TCC. In contrast, the performance gaps in five datasets that CA-TCC beats our method are relatively small. We added these experimental results in the appendix of the revised manuscript.
>
> |  |           CB          |   CX  |   EF  |   L2  |   MS  |   PL  |   PC  |   RD  |   SR  |   SL  |   SC  |   T1  |   TR  |   TP   |   YO  |
> |:------:|:---------------------:|:-----:|:-----:|:-----:|:-----:|:-----:|:-----:|:-----:|:-----:|:-----:|:-----:|:-----:|:-----:|:------:|:-----:|
> | CA-TCC |         **99.84**         | **71.03** | 97.27 | **75.67** | 93.24 | 96.35 | 87.20 | **62.26** | 77.90 | 77.67 | 91.85 | 67.87 | 81.94 | **100.00** | 80.76 |
> |  MDTA  |         99.71         | 64.07 | **99.81** | 75.23 | **95.03** | **96.88** | **93.58** | 59.56 | **99.61** | **86.18** | **98.11** | **94.03** | **98.44** |  99.97 | **85.17** |
>
> [1] Eldele, E., Ragab, M., Chen, Z., Wu, M., Kwoh, C. K., Li, X., & Guan, C. (2023). Self-supervised contrastive representation learning for semi-supervised time-series classification. IEEE Transactions on Pattern Analysis and Machine Intelligence.

---

### Official Review · Reviewer_G4ga · 2023-10-30

**Soundness:** 2 fair
**Presentation:** 3 good
**Contribution:** 2 fair
**Rating:** 3
**Confidence:** 5

**Summary:**

The authors present the concept of masked dual-temporal autoencoders for the semi-supervised classification of time series data. In particular, they introduce a novel loss function known as the relation-preserving loss, which is designed to effectively capture intricate temporal patterns within time series. Empirical assessments conducted on a carefully chosen set of 15 UCR time series datasets conclusively demonstrate that the proposed method attains a performance level that is currently considered state-of-the-art in this domain.

**Strengths:**

1. The authors initially introduced the concept of dual-temporal autoencoders, utilizing a framework rooted in masked time-series modeling, as a foundation for their approach to semi-supervised time-series classification.

2. The authors have incorporated a relation-preserving loss function into their methodology, aiming to enhance the capacity to capture temporal information within time series data in a self-supervised learning manner.

**Weaknesses:**

1. Typically, in semi-supervised learning, it's important to maintain consistency in the employed backbone model for a fair evaluation of different strategies. In this paper, the authors utilized a CNN+Transformer encoder to extract feature representations from time series data, while the baseline methods relied on a four-layer convolutional neural network as their encoder. This discrepancy in backbone architectures raises fairness concerns in the comparison.

2.  The concept of the dual-temporal encoder introduced in this paper lacks novelty. The utilization of a CNN+Transformer encoder for time series feature extraction was previously observed in TS-TCC [R1, R2], which has also been employed for semi-supervised learning of time series data.

3. The relation-preserving loss introduced in this paper shares commonalities with SemiTime [R3]. Both approaches employ binary cross-entropy loss to capture temporal dependencies within time series data. However, the distinction lies in the fact that SemiTime constructs the loss function using two subsequences from the time series, whereas the approach in this paper constructs the loss function using the outputs from a CNN encoder and a Transformer encoder.

4. Some graphs, like Figures 1&6, are too small in size to distinguish different classes.

5. The number of UCR time series datasets utilized for the experiments in this study needs expansion. The creator of the UCR archive [R4], including 128 time series datasets, recommends comprehensive testing and publication of results on all datasets to prevent biased selection unless specific data types are targeted (e.g., classifying short time series). In the context of semi-supervised time series classification, guidelines include not using datasets where the number of instances in the training/validation/test set is smaller than the size of its class labels [R5] and ensuring each category comprises a minimum of 30 samples on average [R6]. Consequently, this study evaluates results on 100 UCR datasets in adherence to [R5], and for [R6], 106 datasets out of the original 128 UCR datasets are utilized for experimental evaluation.

[R1] Time-series representation learning via temporal and contextual contrasting. IJCAI, 2021.

[R2] Self-supervised contrastive representation learning for semi-supervised time-series classification. TPAMI, 2023.

[R3] Semi-supervised time series classification by temporal relation prediction. ICASSP, 2021.

[R4] The UCR time series archive. 2019. https://www.cs.ucr.edu/~eamonn/time_series_data/

[R5] Time-frequency based multi-task learning for semi-supervised time series classification. Information Sciences, 2023.

[R6] Temporal-frequency co-training for time series semi-supervised learning. AAAI, 2023.

**Questions:**

1. In the appendix, the authors write that they used time-warping and magnitude-warping augmentations for all baselines during model training. Why should all baselines use data augmentation? Data augmentation represents merely one facet of semi-supervised learning strategies. If the method presented in this paper were to incorporate the aforementioned data augmentation technique, it would lead to an unfair comparison in experimental outcomes. This is because the primary focus of this study does not centre on data augmentation methods.

2. Setting masking ratio to random seems to be doubtful, the ablation experiment only contains the masking ratio of 0.5.

3. See the above weaknesses.

---

> ### Author Response · Authors · 2023-11-17
> **Response for Reviewer G4ga by Authors**
>
> We appreciate your valuable comments. We tried our best to address all of your comments, including five weaknesses, W1-W5, and two questions, Q1 and Q2. We hope our responses help to clarify our work and answer all the concerns.
>
> **W1** As described in the appendix, we set the encoder architecture of baselines differently from the proposed method because the four-layer convolutional neural network used in previous works [1, 2] performed better than the transformer architecture in most baselines. When we used the transformer as encoder architecture for the baselines, most showed a decreased performance of about 10% on average. Thus, for each strategy, we select the better one between a CNN and a transformer as a backbone network. We presented these results in Table 4.
>
> [1] Xi, L., Yun, Z., Liu, H., Wang, R., Huang, X., & Fan, H. (2022). Semi-supervised time series classification model with self-supervised learning. Engineering Applications of Artificial Intelligence, 116, 105331.
>
> [2] Liu, X., Zhang, F., Liu, H., & Fan, H. (2022). itimes: Investigating semi-supervised time series classification via irregular time sampling. IEEE Transactions on Industrial Informatics.
>
> **W2** The proposed method can be distinguished from TS-TCC [1], namely CA-TCC, in *the architectures and purposes of using CNN and transformer*.
> First, TS-TCC used the Residual Network introduced in [2] as a CNN encoder to extract high-level representations to be augmented for contrastive learning because contrastive learning in the latent space generally performs better than that in the data space [3]. By contrast, the proposed method used CNN with causal padding and dilated filters as the former sub-encoder to reflect various temporal resolutions while enhancing the efficiency of the latter transformer-based sub-encoder [4].
>
> Second, TS-TCC just employed the transformer to extract useful representations used for contextual contrasting. In contrast, the proposed method used the transformer as the encoder architecture of the masked autoencoder for masked time-series modeling, which is effective in capturing fine-grained semantic information of time series.
>
> To summarize, TS-TCC is a contrastive learning-based semi-supervised time-series classification method, which inherits the limitations of the existing contrastive learning methods and transformer architectures: the high sensitivity to data augmentations and the impossibility of considering diverse temporal resolutions. By contrast, we proposed a masked time-series modeling-based semi-supervised time-series classification framework by considering diverse temporal resolutions and random masking ratios. In addition, we did not use contrastive learning, so any data augmentations with strong inductive biases are not required [3].
>
> We added this comparison with TS-TCC in the appendix of the revised manuscript to clarify the differences.
>
> [1] Eldele, E., Ragab, M., Chen, Z., Wu, M., Kwoh, C. K., Li, X., & Guan, C. (2023). Self-supervised contrastive representation learning for semi-supervised time-series classification. IEEE Transactions on Pattern Analysis and Machine Intelligence.
>
> [2] Wang, Z., Yan, W., & Oates, T. (2017, May). Time series classification from scratch with deep neural networks: A strong baseline. In 2017 International joint conference on neural networks (IJCNN) (pp. 1578-1585). IEEE.
>
> [3] Yue, Z., Wang, Y., Duan, J., Yang, T., Huang, C., Tong, Y., & Xu, B. (2022, June). Ts2vec: Towards universal representation of time series. In Proceedings of the AAAI Conference on Artificial Intelligence (Vol. 36, No. 8, pp. 8980-8987).
>
> [4] Fan, A., Lavril, T., Grave, E., Joulin, A., & Sukhbaatar, S. (2020). Addressing some limitations of transformers with feedback memory. arXiv preprint arXiv:2002.09402.
>
> **W3** Our method and SemiTime [1] are totally different regarding *the aims and approaches*, although they share the binary cross-entropy loss.
>
> Our relation-preserving loss aims to prevent the loss of the temporal structural information obtained from the former sub-encoder while passing through the subsequent sub-encoder, whereas the binary cross-entropy used in SemiTime aims to capture temporal dependency by using the past and future segments. Therefore, our relation-preserving loss function minimizes the difference between gram matrices derived from the outputs of two sub-encoders, whereas SemiTime encourages the positive pairs of segments to be consistent and the negative pairs to be distant. To clarify this difference, we added this paragraph in the appendix of the revised manuscript.
>
> [1] Fan, H., Zhang, F., Wang, R., Huang, X., & Li, Z. (2021, June). Semi-supervised time series classification by temporal relation prediction. In ICASSP 2021-2021 IEEE International Conference on Acoustics, Speech and Signal Processing (ICASSP) (pp. 3545-3549). IEEE.

---

> ### Author Response · Authors · 2023-11-17
> **Response for Reviewer G4ga by Authors (cont'd)**
>
> **W4** Due to the page limit, some graphs were displayed small. We added the enlarged Figures 1 and 6 in the appendix to enhance visibility and readability.
>
> **W5** Unfortunately, due to the limited computing resources and time, it was difficult to utilize all datasets in the UCR archive. Therefore, as many previous studies did [1-6], we selected some datasets with various data types, number of data and classes, and sequence lengths. In addition, the number of datasets used in this paper is 15, which is larger than those used in most previous works. We added this paragraph in the appendix.
>
> [1] Jawed, S., Grabocka, J., & Schmidt-Thieme, L. (2020). Self-supervised learning for semi-supervised time series classification. In Advances in Knowledge Discovery and Data Mining: 24th Pacific-Asia Conference, PAKDD 2020, Singapore, May 11–14, 2020, Proceedings, Part I 24 (pp. 499-511). Springer International Publishing.
>
> [2] Fan, H., Zhang, F., Wang, R., Huang, X., & Li, Z. (2021, June). Semi-supervised time series classification by temporal relation prediction. In ICASSP 2021-2021 IEEE International Conference on Acoustics, Speech and Signal Processing (ICASSP) (pp. 3545-3549). IEEE.
>
> [3] Zuo, J., Zeitouni, K., & Taher, Y. (2021, December). Smate: Semi-supervised spatio-temporal representation learning on multivariate time series. In 2021 IEEE International Conference on Data Mining (ICDM) (pp. 1565-1570). IEEE.
>
> [4] Xi, L., Yun, Z., Liu, H., Wang, R., Huang, X., & Fan, H. (2022). Semi-supervised time series classification model with self-supervised learning. Engineering Applications of Artificial Intelligence, 116, 105331.
>
> [5] Liu, X., Zhang, F., Liu, H., & Fan, H. (2022). itimes: Investigating semi-supervised time series classification via irregular time sampling. IEEE Transactions on Industrial Informatics.
>
> [6] Eldele, E., Ragab, M., Chen, Z., Wu, M., Kwoh, C. K., Li, X., & Guan, C. (2023). Self-supervised contrastive representation learning for semi-supervised time-series classification. IEEE Transactions on Pattern Analysis and Machine Intelligence.
>
> **Q1** Unlike the proposed method, the baselines require data augmentations, and the recent works [1, 2] used time-warping and magnitude-warping as augmentation methods in all baselines for fair comparison. Thus, we equally set the data augmentations for baselines by following them.
>
> [1] Xi, L., Yun, Z., Liu, H., Wang, R., Huang, X., & Fan, H. (2022). Semi-supervised time series classification model with self-supervised learning. Engineering Applications of Artificial Intelligence, 116, 105331.
>
> [2] Liu, X., Zhang, F., Liu, H., & Fan, H. (2022). itimes: Investigating semi-supervised time series classification via irregular time sampling. IEEE Transactions on Industrial Informatics.
>
> **Q2** In general, exploring optimal masking ratios for each individual dataset is impractical. In addition, if we consider various masking ratios during model training, the information redundancy originated from the correlation between time steps can be eliminated in diverse perspectives; thereby, a variety of challenging self-supervisory tasks that allow the model to identify sophisticated temporal relations are created. Therefore, we used *random masking ratios* to enhance model performance by identifying intricate temporal relations even without the inefficiency of searching for optimal masking ratios.
>
> As shown in Figure 5, the three fixed masking ratios of 0.2, 0.5, and 0.8 showed notable different performances depending on the datasets. In contrast, even without searching for optimal masking ratios, the random masking ratio achieved the best performance in 12 out of 15 datasets and also showed decent performance in the remaining dataset. In addition, in Table 2, we compared the average classification performance of the random masking ratio with that of the fixed ratio of 0.5 in more detail. Consequently, the average classification performance of the fixed ratio of 0.5 showed a drop rate of 1.81% compared to the proposed random masking ratio. Specifically, as shown in Table 7, which provides the complete results, the performance of the fixed masking ratio of 0.5 remarkably decreased by over 10% in several cases, especially for low label ratios on some datasets. Therefore, we demonstrated that random masking ratios enhance the model’s generalization performance  without the high-cost tuning process for finding optimal making ratios. Furthermore, the effect of random masking ratios is also demonstrated in Figure 7 by showing robustness to missing values while effectively capturing underlying temporal relations.
>
> We added detailed analyses of the experiments for the effect of random masking ratio in the appendix of the revised manuscript to emphasize the importance of random masking ratios.

---

### Official Review · Reviewer_6QXJ · 2023-11-01

**Soundness:** 3 good
**Presentation:** 3 good
**Contribution:** 3 good
**Rating:** 6
**Confidence:** 4

**Summary:**

This paper proposes a novel framework named masked dual-temporal autoencoders (MDTA) for semi-supervised time-series classification. MDTA is the first masked time-series modeling framework for semi-supervised time-series classification. MDTA could captures relevant semantic information from unlabeled time series and incorporates it with supervisory features obtained from labeled ones to enhance model performance. Also random masking ratios during traing makes MDTA avoid the high-cost tuning process for finding optimal making ratios. The superiority of MDTA over baseline approaches is demonstrated by extensive comparative experiments.

**Strengths:**

1. MDTA is the first masked time-series modeling framework for semi-supervised time-series classification.
2. And in fact, the paper's idea of predicting in unlabeled time series subsets to extract features and using the extracted features for classification is novel.
3. The writing and the overall logical arrangement of this paper are reasonable, so that readers can grasp the key points.

**Weaknesses:**

1. The biggest drawback of this article is motivation. From the expression of the article, motivation is that MTM has not been used to solve semi supervised time series classification.
2. In Section 4.1, the text "Figure 5 and Table 2 demonstrate that random masking ratios enhance classification performance without the high-cost tuning process for finding optimal making ratios." is not rigorous enough. Figure 5 and Table 2 can only demonstrate that random masking ratios performs better than fixed making ratios, as the experimental masking ratios 0.2, 0.5, and 0.8 may not necessarily be optimal making ratios for every dataset.
3. The superscript of unlabeled set in Section 3.1 should be n_u instead of n.

**Questions:**

1. What is the proportion of unlabeled and labeled data in each dataset?
2. From the model diagram Figure 3(a) alone, is the output one-dimensional? If it is one-dimensional, it does not match the description.

---

> ### Author Response · Authors · 2023-11-17
> **Response for Reviewer 6QXJ by Authors**
>
> We appreciate your helpful comments and rating. We tried our best to address all of your comments, including three weaknesses, W1-W3, and two questions, Q1 and Q2. We hope our responses help to clarify our work and answer all the concerns.
>
> **W1** Previous works for semi-supervised time-series classification have exploited self-supervised learning, including contrastive learning, to learn underlying structures within unlabeled time series [1-3]. These approaches often capture coarse-grained context information focused on the instance level, insufficiently recognizing temporal patterns of time series [4]. In addition, the model performance highly depends on techniques to construct self-generated labels [5, 6]. Thus, some recent works [7-9] introduced masked time-series modeling (MTM), which solves high sensitivity to constructing self-generated labels and captures fine-grained context information. However, MTM has not yet been introduced to semi-supervised time-series classification. Also, directly applying the existing MTMs to semi-supervised time-series classification still has two potential limitations: *the impossibility of reflecting diverse temporal resolutions* and *high sensitivity to masking ratios*. Since temporal dependencies span various time intervals, reflecting diverse resolutions can significantly improve model performance [7]. In addition, if we use several masking ratios during model training together, we can eliminate the information redundancy differently; thereby, a variety of challenging self-supervisory tasks that allow the model to identify sophisticated temporal relations are created. Thus, we proposed the first MTM-based framework suitable for semi-supervised time-series classification, which effectively considers various temporal resolutions by dual-temporal encoder while mitigating sensitivity to masking ratios through random masking ratios. We revised Section 1 to highlight this paragraph that can clarify the motivation of our work.
>
> [1] Fan, H., Zhang, F., Wang, R., Huang, X., & Li, Z. (2021, June). Semi-supervised time series classification by temporal relation prediction. In ICASSP 2021-2021 IEEE International Conference on Acoustics, Speech and Signal Processing (ICASSP) (pp. 3545-3549). IEEE.
>
> [2] Xi, L., Yun, Z., Liu, H., Wang, R., Huang, X., & Fan, H. (2022). Semi-supervised time series classification model with self-supervised learning. Engineering Applications of Artificial Intelligence, 116, 105331.
>
> [3] Liu, X., Zhang, F., Liu, H., & Fan, H. (2022). itimes: Investigating semi-supervised time series classification via irregular time sampling. IEEE Transactions on Industrial Informatics.
>
> [4] Wang, T., & Isola, P. (2020, November). Understanding contrastive representation learning through alignment and uniformity on the hypersphere. In International Conference on Machine Learning (pp. 9929-9939). PMLR.
>
> [5] You, Y., Chen, T., Shen, Y., & Wang, Z. (2021, July). Graph contrastive learning automated. In International Conference on Machine Learning (pp. 12121-12132). PMLR.
>
> [6] Yue, Z., Wang, Y., Duan, J., Yang, T., Huang, C., Tong, Y., & Xu, B. (2022, June). Ts2vec: Towards universal representation of time series. In Proceedings of the AAAI Conference on Artificial Intelligence (Vol. 36, No. 8, pp. 8980-8987).
>
> [7] Zerveas, G., Jayaraman, S., Patel, D., Bhamidipaty, A., & Eickhoff, C. (2021, August). A transformer-based framework for multivariate time series representation learning. In Proceedings of the 27th ACM SIGKDD conference on knowledge discovery & data mining (pp. 2114-2124).
>
> [8] Nie, Y., Nguyen, N. H., Sinthong, P., & Kalagnanam, J. (2022). A time series is worth 64 words: Long-term forecasting with transformers. arXiv preprint arXiv:2211.14730.
>
> [9] Dong, J., Wu, H., Zhang, H., Zhang, L., Wang, J., & Long, M. (2023). SimMTM: A Simple Pre-Training Framework for Masked Time-Series Modeling. arXiv preprint arXiv:2302.00861.
>
> **W2** Through Figure 5 and Table 2, we aimed to demonstrate the effect of random masking ratios that enhance the model's generalization performance by identifying intricate temporal relations even without the inefficiency of searching for proper masking ratios. Thus, we compared the random masking ratio with three fixed masking ratios of 0.2 (low), 0.5 (medium), and 0.8 (high). Although these fixed masking ratios may not be optimal for every dataset, we can examine the overall tendency of each dataset against the low, medium, and high masking ratios. Through the results, we demonstrated the random masking ratio enhances the generalization performance of the model without the high-cost tuning process for finding optimal masking ratios. We added this paragraph in the appendix of the revised manuscript. In addition, we modified the sentence mentioned above as follows:
> - Figure 5 and Table 2 demonstrate that random masking ratios enhance the model’s generalization performance without the high-cost tuning process for finding optimal making ratios.

---

> ### Author Response · Authors · 2023-11-17
> **Response for Reviewer 6QXJ by Authors (cont'd)**
>
> **W3** The superscript of unlabeled set in Section 3.1 is $n$. Since $\mathbb{D}=\mathbb{D}\_{\ell} \cup \mathbb{D}\_{u}$, where the sizes of $\mathbb{D}\_{\ell}$ and $\mathbb{D}\_{u}$ are $n\_{\ell}$ and $n\_{u}=n-n\_{\ell}$, respectively, the indices of labeled instances are $1,2,⋯,n\_{\ell}$, and those of unlabeled ones are $n\_{\ell}+1,n\_{\ell}+2,⋯,n$; thereby, the unlabeled set $\mathbb{D}\_{u}=\\{(\boldsymbol{x}\_i,\cdot)\\}\_{(i=n\_{\ell}+1)}^n$.
>
> **Q1** We evaluated the model performance on semi-supervised time-series classification with label ratios $\in \\{0.1,0.2,⋯,0.9\\}$. In Table 1, we provided the average accuracy scores across label ratios from 0.1 to 0.9 on each dataset. The complete results with standard deviations for all label ratios $\in \\{0.1,0.2,⋯,0.9\\}$ are given in Table 5.
>
> **Q2** The output $u\_i \in \mathbb{R}^{t \times d\_u}$ of the sub-encoder $f\_1$ is not one-dimensional. $u\_i$ has $d\_u$-dimensional features over the sequence length $t$. The expression ‘temporal features’ may confuse the dimension of $u\_i$. We modified the ‘temporal features’ expressed in Figure 3(a) as ‘temporal representation’ in the revised manuscript.

---

### Official Review · Reviewer_8ZuZ · 2023-11-08

**Soundness:** 3 good
**Presentation:** 2 fair
**Contribution:** 3 good
**Rating:** 6
**Confidence:** 4

**Summary:**

This study introduces a novel semi-supervised time series classification framework based on Masked Time Series Modeling (MTM). MTM effectively captures complex temporal patterns by combining supervised features with semantic information extracted from unlabeled time series. The method employs a Dual Temporal Autoencoder with a relation-preserving loss function, random masking ratios, and shows superior performance in semi-supervised time-series classification compared to state-of-the-art methods. It addresses challenges in label sparsity and sensitivity to self-generated labels, providing a promising approach for leveraging unlabeled time-series data.

The paper introduces a new framework for semi-supervised time series classification based on masked time series modeling. The authors provide a good description of their motivation and the proposed method seems to address the targeted problem. More investigation about some designs and parameters should be analyzed.

**Strengths:**

1. Overall, this paper is well-written and easy to follow.
2. This paper proposes a novel semi-supervised time series classification method based on masked time-series modeling, which combines semantic information in unlabeled time series with supervised information in labeled time series to improve classification performance. This is the first work to introduce masked time-series modeling to the task of semi-supervised time-series classification.
3. Through extensive experimental results, this method is significantly better than the SOTA methods in semi-supervised time series classification tasks.

**Weaknesses:**

1.Important and basic information about baselines and experimental settings should be placed in the main text, not in the appendix.

2. Random masking ratios is an important component of the method. However, according to Figure 5 and Table 2, when random masking ratios is replaced by different fixed masking ratios, MDTA exhibits the same or lower performance than some fixed masking ratios on some datasets, which makes it difficult to convince of the effect of random masking ratios. Although the computation cost of exploration is reduced, the gain from random masking ratios on each dataset cannot be guaranteed. So, the overall effect of MDTA may be less contributed by random masking ratios. Is there a better explanation or improvement?

3. There is a lack of investigation on the impacts of hyper-parameters, such as $\alpha$ and $\beta$.

4. Transformer-based sub-encoder $f_2$ is a key component in MDTA. The experiments do not compare with some SOTA transformer models.

**Questions:**

See weaknesses

---

> ### Author Response · Authors · 2023-11-17
> **Response for Reviewer 8ZuZ by Authors**
>
> We appreciate your valuable comments. We tried our best to address all of your comments, including four weaknesses, W1-W4. We hope our responses help to clarify our work.
>
> **W1** Due to the page limit, we provided all experimental settings and implementation details in the appendix. In the revised version, we briefly mentioned the experimental settings in Section 4.
>
> **W2** In general, exploring optimal masking ratios for each individual dataset within reasonable time and cost is impractical. In addition, if we consider various masking ratios during model training, the information redundancy originating from the correlation between time steps can be eliminated in diverse perspectives; thereby, a variety of challenging self-supervisory tasks that allow the model to identify sophisticated temporal relations are created. Therefore, we used random masking ratios to enhance model performance by identifying intricate temporal relations without the inefficiency of searching for proper masking ratios.
>
> As shown in Figure 5, when we fix the masking ratio to a certain value, the performances highly vary by the datasets. In contrast, the random masking ratio achieved the best performance in 12 out of 15 datasets and also showed decent performances in the remaining datasets. Moreover, as shown in Table 7, which provides the complete results, the performance of the fixed masking ratio remarkably decreased by over 10% in several cases, especially low label ratios on some datasets. Furthermore, the effect of random masking ratios is also demonstrated in Figure 7 by showing robustness to missing values as well as effectively capturing underlying temporal relations.
>
> We added detailed analyses of the experiments for the effect of random masking ratio to emphasize its importance in the appendix of the revised manuscript.
>
> **W3** To examine the impact of loss weights, $\alpha$ and $\beta$, used in equation (9) for the masked time-series modeling paradigm, we performed sensitivity analysis against them. We compared the adaptive loss weighting strategy [1] to three fixed loss weights. Here, the pairs of $\alpha$ and $\beta$ for the fixed loss weights were set to (0.2, 0.8), (0.5, 0.5), and (0.8, 0.2). Consequently, as shown in the *Figure 9 of the revised manuscript*, the adaptive loss weighting strategy exhibits comparable performance with the three fixed loss weights on average. However, in some datasets, such as CricketX, SwedishLeaf, and Yoga, the adaptive loss weighting performs notably better than the others. In addition, it allows us to reduce the effort for finding optimal values for $\alpha$ and $\beta$ on each dataset. Therefore, we used the adaptive loss weighting strategy for the experiments performed in this paper. In the revised manuscript, we moved the description for $\alpha$ and $\beta$ from Section 3 to Section B and added this sensitivity analysis in Section H of the appendix.
>
> [1] Heydari, A. A., Thompson, C. A., & Mehmood, A. (2019). Softadapt: Techniques for adaptive loss weighting of neural networks with multi-part loss functions. arXiv preprint arXiv:1912.12355.
>
> **W4** The proposed method effectively reflects diverse temporal resolutions by the dual-temporal encoder architecture and mitigates sensitivity to masking ratios by random masking ratios. In this work, we mainly focus on the effects of (1) the dual-temporal autoencoder framework to reflect diverse temporal resolutions, (2) the random masking ratio that can reduce sensitivity to the masking ratio selection, and (3) the relation preserving loss that prevents information loss in a deep transformer architecture. Thus, we employed a simple transformer introduced in [1] for our proposed method.
> However, our transformer-based sub-encoder $f_2$ can be flexibly replaced by any model with a similar transformer architecture. In other words, we agree that using the SOTA transformers may improve the performance of the proposed framework.
>
> In Section 3.2.1 of the revised manuscript, we mentioned the flexibility of $f_2$.
>
> [1] Zerveas, G., Jayaraman, S., Patel, D., Bhamidipaty, A., & Eickhoff, C. (2021, August). A transformer-based framework for multivariate time series representation learning. In Proceedings of the 27th ACM SIGKDD conference on knowledge discovery & data mining (pp. 2114-2124).

---

### Author Response · Authors · 2023-11-17
**Revised manuscript updated**

Dear Reviewers,

We, the authors, appreciate your thorough reading and great comments.

Based on the comments, we improved our manuscript and uploaded it as a revision.

All of the changes made in the revised manuscript have been marked in *YELLOW*.

Please check our revised manuscript as well as the responses to your comments.

Sincerely,

Authors

---

### Meta-Review · Area_Chair_LhrS · 2023-12-10

**Metareview:**

The authors present masked dual-temporal autoencoder for semi-supervised time series classification. In particular, the proposed relation-preserving loss is novel and can capture the intricate temporal patterns within the time series data. Overall, this paper has some good technical contributions but its current form still falls below the ICLR standard. The reviewers have severe concerns about the evaluation. The experiments in this paper deviate from the standard setting of semi-supervised classification. The authors need to ensure consistency in both the backbone architecture and the training strategy during evaluation. In addition, the experimental part also lacks in-depth analysis. For these reasons, my recommendation is to reject this paper.

**Justification For Why Not Higher Score:**

The experimental settings (e.g., backbone) in this paper are not standard. The paper also lacks in-depth analysis for the results. The evaluation in this paper is not convincing due to the above reasons, and thus I would like to reject this paper.

**Justification For Why Not Lower Score:**

N/A.

---

### Decision · Program_Chairs · 2024-01-16

Reject